# Amortized Inference for Causal Structure Learning

**Lars Lorch**
ETH Zurich
Zurich, Switzerland
`llorch@ethz.ch`

**Scott Sussex**
ETH Zurich
Zurich, Switzerland
`ssussex@ethz.ch`

**Jonas Rothfuss**
ETH Zurich
Zurich, Switzerland
`rojonas@ethz.ch`

**Andreas Krause**[*]
ETH Zurich
Zurich, Switzerland
`krausea@ethz.ch`

**Bernhard Schölkopf**[*]
MPI for Intelligent Systems
Tübingen, Germany
`bs@tuebingen.mpg.de`

## Abstract

Inferring causal structure poses a combinatorial search problem that typically involves evaluating structures with a score or independence test. The resulting search is costly, and designing suitable scores or tests that capture prior knowledge is difficult. In this work, we propose to *amortize causal structure learning*. Rather than searching over structures, we train a variational inference model to directly predict the causal structure from observational or interventional data. This allows our inference model to acquire domain-specific inductive biases for causal discovery solely from data generated by a simulator, bypassing both the hand-engineering of suitable score functions and the search over graphs. The architecture of our inference model emulates permutation invariances that are crucial for statistical efficiency in structure learning, which facilitates generalization to significantly larger problem instances than seen during training. On synthetic data and semisynthetic gene expression data, our models exhibit robust generalization capabilities when subject to substantial distribution shifts and significantly outperform existing algorithms, especially in the challenging genomics domain. Our code and models are publicly available at: `https://github.com/larslorch/avici`.

## 1 Introduction

Learning the causal structure among a set of variables is a fundamental task in various scientific disciplines (Spirtes et al., 2000; Pearl, 2009). However, inferring this causal structure from observations of the variables is a difficult inverse problem. The solution space of potential causal structures, usually modeled as directed graphs, grows superexponentially with the number of variables. To infer a causal structure, standard methods have to search over potential graphs, usually maximizing either a graph scoring function or testing for conditional independences (Heinze-Deml et al., 2018).

Specifying realistic inductive biases is universally difficult for existing approaches to causal discovery. Score-based methods use strong assumptions about the data-generating process, such as linearity (Shimizu et al., 2006), specific noise models (Hoyer et al., 2008; Peters and Bühlmann, 2014), and the absence of measurement error (cf. Scheines and Ramsey 2016; Zhang et al. 2017), which are difficult to verify (Dawid, 2010; Reisach et al., 2021). Conversely, constraint-based methods do not have enough domain-specific inductive bias. Even with an arbitrarily large dataset, they are limited to identifying equivalence classes that may be exponentially large (He et al., 2015b). Moreover, the search over directed graphs itself may introduce unwanted bias and artifacts (cf. Colombo et al. 2014).

36th Conference on Neural Information Processing Systems (NeurIPS 2022).

---

[*]Equal supervision.

The intractable search space ultimately imposes hard constraints on the causal structure, e.g., the node degree (Spirtes et al., 2000), which limits the suitability of search in real-world domains.

In the present work, we propose to *amortize* causal structure learning. In other words, our goal is to optimize an inference model to directly predict a causal structure from a provided dataset. We show that this approach allows inferring causal structure solely based on synthetic data generated by a *simulator* of the data-generating process we are interested in. Much effort in the sciences, for example, goes into the development of realistic simulators for high-impact and yet challenging causal discovery domains, like gene regulatory networks (Schaffter et al., 2011; Dibaeinia and Sinha, 2020), fMRI brain responses (Buxton, 2009; Bassett and Sporns, 2017), and chemical kinetics (Anderson and Kurtz, 2011; Wilkinson, 2018). Our approach based on amortized variational inference (AVICI) ultimately allows us to both specify domain-specific inductive biases not easily represented by graph scoring functions and bypass the problems of structure search. Our model architecture is permutation in- and equivariant with respect to the observation and variable dimensions of the provided dataset, respectively, and generalizes to significantly larger problem instances than seen during training.

On synthetic data and semisynthetic gene expression data, our approach significantly outperforms existing algorithms for causal discovery, often by a large margin. Moreover, we demonstrate that our inference models induce calibrated uncertainties and robust behavior when subject to substantial distribution shifts of graphs, mechanisms, noise, and problem sizes. This suggests that our pretrained models are not only fast but also both reliable and versatile for future downstream use. In particular, AVICI was the only method to infer plausible causal structures from noisy gene expression data, advancing the frontiers of structure discovery in fields such as biology.

## 2   Background and Related Work

### 2.1   Causal Structure

In this work, we follow Mooij et al. (2016) and define the causal structure $G$ of a set of $d$ variables $\mathbf{x} = (x_1, \dots, x_d)$ as the directed graph over $\mathbf{x}$ whose edges represent all *direct causal* effects among the variables. A variable $x_i$ has a direct causal effect on $x_j$ if intervening on $x_i$ affects the outcome of $x_j$ independent of the other variables $\mathbf{x}_{\setminus ij} := \mathbf{x} \setminus \{x_i, x_j\}$, i.e., there exists $a \neq a'$ such that

$$p(x_j \,|\, \mathrm{do}(x_i = a, \mathbf{x}_{\setminus ij} = \mathbf{c})) \neq p(x_j \,|\, \mathrm{do}(x_i = a', \mathbf{x}_{\setminus ij} = \mathbf{c})) \tag{1}$$

for some $\mathbf{c}$. An *intervention* $\mathrm{do}(\cdot)$ denotes any active manipulation of the generative process of $\mathbf{x}$, like gene knockouts, in which the transcription rates of genes are externally set to zero. Other models such as causal Bayesian networks and structural causal models (Peters et al., 2017) are less well-suited for describing systems with feedback loops, which we consider practically relevant. However, we note that our approach does not require any particular formalization of causal structure. In particular, we later show how to apply our approach when $G$ is constrained to be acyclic. We assume causal sufficiency, i.e., that $\mathbf{x}$ contains all common causal parents of the variables $x_i$ (Peters et al., 2017).

### 2.2   Related Work

Classical methods for causal structure learning search over causal graphs and evaluate them using a likelihood or conditional independence test (Chickering, 2003; Kalisch and Bühlman, 2007; Hauser and Bühlmann, 2012; Zheng et al., 2018; Heinze-Deml et al., 2018). Other methods combine constraint- and score-based ideas (Tsamardinos et al., 2006) or use the noise properties of an SCM that is postulated to underlie the data-generating process (Shimizu et al., 2006; Hoyer et al., 2008).

Deep learning has been used for causal inference, e.g., for estimating treatment effects (Shalit et al., 2017; Louizos et al., 2017; Yoon et al., 2018) and in instrumental variable analysis (Hartford et al., 2017; Bennett et al., 2019). In structure learning, neural networks have primarily been used to model nonlinear causal mechanisms (Goudet et al., 2018; Yu et al., 2019; Lachapelle et al., 2020; Brouillard et al., 2020; Lorch et al., 2021) or to infer the structure of a single dataset (Zhu et al., 2020). Prior work applying amortized inference to causal discovery only studied narrowly defined subproblems such as the bivariate case (Lopez-Paz et al., 2015) and fixed causal mechanisms (Löwe et al., 2022) or used correlation coefficients for prediction (Li et al., 2020). In concurrent work, Ke et al. (2022) also frame causal discovery as supervised learning, but with significant differences. Most importantly, we optimize a variational objective under a model class that captures the symmetries of structure learning. Empirically, our models generalize to much larger problem sizes, even on realistic genomics data.

# 3 AVICI: Amortized Variational Inference for Causal Discovery

## 3.1 Variational Objective

To amortize causal structure learning, we define a data-generating distribution $p(D)$ that models the domain in which we infer causal structures. The observations $D = \{\mathbf{x}^1, \ldots, \mathbf{x}^n\} \sim p(D)$ are generated by sampling from a distribution over causal structures $p(G)$ and then obtaining realizations from a data-generating mechanism $p(D \mid G)$. The data-generating process $p(D \mid G)$ characterizes all direct causal effects (1) in the system, but it is not necessarily induced by ancestral sampling over a directed acyclic graph. Real-world systems are often more naturally modeled at different granularities or as dynamical systems (Mooij et al., 2013; Hoel et al., 2013; Rubenstein et al., 2017; Schölkopf, 2019).

Given a set of observations $D$, our goal is to approximate the posterior over causal structures $p(G \mid D)$ with a variational distribution $q(G; \theta)$. To amortize this inference task for the domain distribution $p(D)$, we optimize an inference model $f_\phi$ to predict the variational parameters $\theta$ by minimizing the expected *forward KL divergence* from the intractable posterior $p(G \mid D)$ to $q(G; \theta)$ for $D \sim p(D)$:

$$\min_\phi \; \mathbb{E}_{p(D)} \, D_{KL}\big(p(G \mid D) \big\| q(G; f_\phi(D))\big) \tag{2}$$

Since it is not tractable to compute the true posterior in (2), we make use of ideas by Barber and Agakov (2004) and rewrite the expected forward KL to obtain an equivalent, tractable objective:

$$\mathbb{E}_{p(D)} \, D_{KL}\big(p(G \mid D) \big\| q(G; f_\phi(D))\big) = \mathbb{E}_{p(D)} \, \mathbb{E}_{p(G \mid D)}[\log p(G \mid D) - \log q(G; f_\phi(D))]$$
$$= -\mathbb{E}_{p(G)} \, \mathbb{E}_{p(D \mid G)}[\log q(G; f_\phi(D))] + \text{const.} \tag{3}$$

The constant does not depend on $\phi$, so we can maximize $\mathcal{L}(\phi) := \mathbb{E}_{p(G)} \, \mathbb{E}_{p(D \mid G)}[\log q(G; f_\phi(D))]$, which allows us to perform amortized variational inference for causal discovery (AVICI). While the domain distribution $p(D) = \mathbb{E}_{p(G)}[p(D \mid G)]$ can be arbitrarily complex, $\mathcal{L}$ is tractable whenever we have access to the causal graph $G$ underlying the generative process of $D$, i.e., to samples from the joint distribution $p(G, D)$. In practice, $p(G)$ and $p(D \mid G)$ can thus be specified by a simulator.

From an information-theoretic viewpoint, the objective (2) maximizes a variational lower bound on the mutual information $I[G; D]$ between the causal structure $G$ and the observations $D$ (Barber and Agakov, 2004). Starting from the definition of mutual information, we obtain

$$I[G; D] = H[G] - H[G \mid D] = H[G] + \mathbb{E}_{p(G,D)}[\log p(G \mid D)]$$
$$\geq H[G] + \mathbb{E}_{p(G,D)}[\log q(G; f_\phi(D))] = H[G] + \mathcal{L}(\phi) \tag{4}$$

where the entropy $H[G]$ is constant. The bound is tight if $\mathbb{E}_{p(D)} D_{KL}(p(G \mid D) \| q(G; f_\phi(D))) = 0$.

## 3.2 Likelihood-Free Inference using the Forward KL

The AVICI objective in (3) intentionally targets the forward KL $D_{KL}(p \, \| \, q(\,\cdot\,; \theta))$, which requires optimizing $\mathbb{E}_{p(G,D)}[\log q(G; \theta)]$. This choice implies that we both model the density $q(G; \theta)$ explicitly and assume access to *samples* from the true data-generating distribution $p(G, D)$. Minimizing the forward KL enables us to infer causal structures in arbitrarily complex domains—that is, even domains where it is difficult to specify an explicit likelihood $p(D \mid G)$. Moreover, the forward KL typically yields more reliable uncertainty estimates since it does not suffer from the variance underestimation problems common to the reverse KL (Bishop and Nasrabadi, 2006).

In contrast, variational inference usually optimizes the reverse KL $D_{KL}(q \, \| \, p)$, which involves the reconstruction term $\mathbb{E}_{q(G;\theta)}[\log p(D \mid G)]$ (Blei et al., 2017). This objective requires a tractable marginal likelihood $p(D \mid G)$. Unless inferring the mechanism parameters jointly (e.g. Brouillard et al. 2020; Lorch et al. 2021), this requirement limits inference to conjugate models with linear Gaussian or categorical mechanisms that assume zero measurement error (Geiger and Heckerman, 1994; Heckerman et al., 1995), which are not justified in practice (Friston et al., 2000; Schaffter et al., 2011; Runge et al., 2019; Dibaeinia and Sinha, 2020). Furthermore, unless the noise scale is learned jointly, likelihoods can be sensitive to the measurement scale of $\mathbf{x}$ (Reisach et al., 2021).

# 4 Inference Model

In the following section, we describe a choice for the variational distribution $q(G; \theta)$ and the inference model $f_\phi$ that predicts $\theta$ given $D$. After that, we detail our training procedure for optimizing the model parameters $\phi$ and for learning causal graphs with acyclicity constraints.

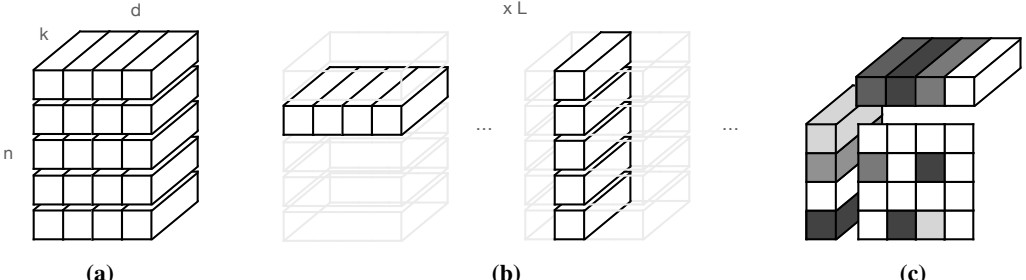

**Figure 1: Model architecture.** **(a)** Our model maps the input to a three dimensional tensor of shape $n \times d \times k$ and remains permutation in- and equivariant over axes $n$ and $d$, respectively. **(b)** Each of the $L$ layers first self-attends over axis $d$ and then over $n$, sharing parameters across the other axis. **(c)** The inner product of two variables' representations models the probability of a direct causal effect.

### 4.1 Variational Family

While any inference model that defines a density is feasible for maximizing the objective in (3), we opt to use a factorized variational family in this work.

$$q(G; \theta) = \prod_{i,j} q(g_{i,j}; \theta_{i,j}) \text{ with } g_{i,j} \sim \text{Bern}(\theta_{i,j}) \tag{5}$$

The inference model $f_\phi$ maps a dataset $D$ corresponding to $n$ samples $\{\boldsymbol{o}^1, \dots, \boldsymbol{o}^n\}$ to a $d$-by-$d$ matrix $\theta$ parameterizing the variational approximation of the causal graph posterior. In addition to the joint observation $\mathbf{x}^i = (x_1^i, \dots, x_d^i)$, each sample $\boldsymbol{o}^i = (o_1^i, \dots, o_d^i)$ may contain interventional information for each variable. When interventions or gene knockouts are performed, we set $o_j^i = (x_j^i, u_j^i)$ and $u_j^i \in \{0, 1\}$ indicating whether variable $j$ was intervened upon in sample $i$. Other settings could be encoded analogously, e.g., when the intervention targets are unknown or measurements incomplete.

### 4.2 Model Architecture

To maximize statistical efficiency, $f_\phi$ should satisfy the symmetries inherent to the task of causal structure learning. Firstly, $f_\phi$ should be *permutation invariant* across the sample dimension (axis $n$). Shuffling the samples should not influence the prediction, i.e., for any permutation $\pi$, we have $f_\phi(\pi(\{\boldsymbol{o}\})) = f_\phi(\{\boldsymbol{o}\})$. Moreover, $f_\phi$ should be *permutation equivariant* across the variable dimension (axis $d$). Reordering the variables should permute the predicted causal edge probabilities, i.e., $f_\phi(\{\boldsymbol{o}_{\pi(1:d)}\})_{i,j} = f_\phi(\{\boldsymbol{o}_{1:d}\})_{\pi(i),\pi(j)}$. Lastly, $f_\phi$ should apply to any $d, n \geq 1$.

In the following, we show how to parameterize $f_\phi$ as a neural network that encodes these properties. After first mapping each $o_j^i$ to a real-valued vector using a position-wise linear layer, $f_\phi$ operates over a continuous, three-dimensional tensor of $n$ rows for the observations, $d$ columns for the variables, and feature size $k$. Figure 1 illustrates the key components of the architecture.

**Attending over axes $d$ and $n$**  The core of $f_\phi$ is composed of $L = 8$ identical layers. Each layer consists of four residual sublayers, where the first and third apply multi-head self-attention and the second and fourth position-wise feed-forward networks, similar to the Transformer encoder (Vaswani et al., 2017). To enable information flow across all $n \times d$ tokens of the representation, the model alternates in attending over the observation and the variable dimension (Kossen et al., 2021). Specifically, the first self-attention sublayer attends over axis $d$, treating axis $n$ as a batch dimension; the second attends over axis $n$, treating axis $d$ as a batch dimension. Since modules are shared across non-attended axes, the representation is permutation equivariant over axes $n$ and $d$ at all times (Lee et al., 2019b).

**Variational parameters**  After building up a representation tensor from the input using the attention layers, we max-pool over the observation axis $n$ to obtain a representation $(\mathbf{z}^1, \dots, \mathbf{z}^d)$ consisting of one vector $\mathbf{z}^i \in \mathbb{R}^k$ for each causal variable. Following Lorch et al. (2021), we use two position-wise linear layers to map each $\mathbf{z}^i$ to two embeddings $\mathbf{u}^i, \mathbf{v}^i \in \mathbb{R}^k$, which are $\ell_2$ normalized. We then model the probability of each edge in the causal graph with an inner product:

$$\theta_{i,j} = \sigma\big(\tau \, \mathbf{u}^i \cdot \mathbf{v}^j + b\big) \tag{6}$$

where $\sigma$ is the logistic function, $b$ a learned bias, and $\tau$ a positive scale that is learned in $\log$ space. Since max-pooling is invariant to permutations and since (6) permutes with respect to axis $d$, $f_\phi$ satisfies the required permutation invariance over axis $n$ and permutation equivariance over axis $d$.

## 4.3 Acyclicity

Cyclic causal effects often occur, e.g., when modeling stationary distributions of dynamical systems, and thus loops in a causal structure are possible. However, certain domains may be more accurately modeled by acyclic structures (Rubenstein et al., 2017). While the variational family in (5) cannot enforce it, we can optimize for acyclicity through $\phi$. Whenever the acyclicity prior is justified, we amend the optimization problem in (2) with the constraint that $q$ only models acyclic graphs in expectation:

$$\mathcal{F}(\phi) := \mathbb{E}_{p(D)}\left[h(f_\phi(D))\right] = 0 \tag{7}$$

The function $h$ is zero if and only if the predicted edge probabilities induce an acyclic graph. We use the insight by Lee et al. (2019a), who show that acyclicity is equivalent to the spectral radius $\rho$, i.e., the largest absolute eigenvalue, of the predicted matrix being zero. We use power iteration to approximate and differentiate through the largest eigenvalue of $f_\phi(D)$ (Golub and Van der Vorst, 2000; Lee et al., 2019a):

$$h(W) := \rho(W) \approx \frac{\mathbf{a}^\top W \mathbf{b}}{\mathbf{a}^\top \mathbf{b}} \quad \text{where for } t \text{ steps:} \quad \begin{array}{l} \mathbf{a} \leftarrow \mathbf{a}^\top W \,/\, \|\mathbf{a}^\top W\|_2 \\ \mathbf{b} \leftarrow W\mathbf{b} \,/\, \|W\mathbf{b}\|_2 \end{array} \tag{8}$$

and $\mathbf{a}, \mathbf{b} \in \mathbb{R}^d$ are initialized randomly. Since a few steps $t$ are sufficient in practice, (8) scales with $O(d^2)$ and is significantly more efficient than $O(d^3)$ constraints based on matrix powers (Zheng et al., 2018; Yu et al., 2019). We do not backpropagate gradients with respect to $\phi$ through $\mathbf{a}, \mathbf{b}$.

## 4.4 Optimization

Combining the objective in (3) with our inference model (5), we can directly use stochastic optimization to train the parameters $\phi$ of the inference model. The expectations over $p(G, D)$ inside $\mathcal{L}$ and $\mathcal{F}$ are approximated using samples from the data-generating process of the domain. When enforcing acyclicity, causal discovery algorithms often use the augmented Lagrangian method for constrained

---

**Algorithm 1** Training the inference model $f_\phi$

---

Parameters: $\phi$ variational, $\lambda$ dual, $\eta$ step size
**while** not converged **do**
    **for** $l$ **steps do**
        $\Delta\phi \propto \nabla_\phi\big(\mathcal{L}(\phi) - \lambda\mathcal{F}(\phi)\big)$
    $\lambda \leftarrow \lambda + \eta\mathcal{F}(\phi)$

---

optimization (e.g., Zheng et al. 2018; Brouillard et al. 2020). In this work, we optimize the parameters $\phi$ of a neural network, so we rely on methods specifically tailored for deep learning and solve the constrained program $\max_\phi \mathcal{L}(\phi)$ s.t. $\mathcal{F}(\phi) = 0$ through its dual formulation (Nandwani et al., 2019):

$$\min_\lambda \max_\phi \; \mathcal{L}(\phi) - \lambda\mathcal{F}(\phi) \tag{9}$$

Algorithm 1 summarizes the general optimization procedure for $q_\phi$, which converges to a local optimum under regularity conditions on the learning rates (Jin et al., 2020). Without an acyclicity constraint, training reduces to the primal updates of $\phi$ with $\lambda = 0$.

## 5 Experimental Setup

Evaluating causal discovery algorithms is difficult since there are few interesting real-world datasets that come with ground-truth causal structure. Often, the believed ground truths may be incomplete or change as expert knowledge improves (Schaffter et al., 2011; Mooij et al., 2020). Following prior work, we deal with this difficulty by evaluating our approach using simulated data with known causal structure and by controlling for various aspects of the task. In Appendix E, we additionally report results on a real-world proteomics dataset (Sachs et al., 2005).

### 5.1 Domains and Simulated Components

We study three domains: two classes of structural causal models (SCMs) as well as semisynthetic single-cell expression data of gene regulatory networks (GRNs). To study the generalization of AVICI beyond the training distribution $p(D)$, we carefully construct a spectrum of test distributions $\tilde{p}(D)$ that incur substantial shift from $p(D)$ in terms of the causal structures, mechanisms, and noise, which we study in various combinations. Whenever we consider interventional data in our experiments, half of the dataset consists of observational data and half of single-variable interventions.

**Data-generating processes** $p(D \,|\, G)$    We consider SCMs with linear functions (LINEAR) and with nonlinear functions of random Fourier features (RFF) that correspond to functions drawn

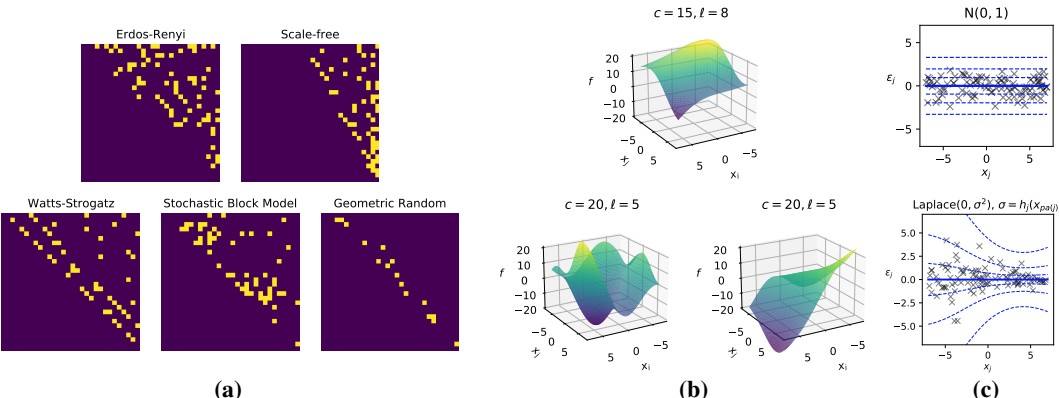

**Figure 2: Moving out-of-distribution in the RFF domain.** Randomly sampled data-generating components of the nonlinear SCM domain during training $p(D)$ (top) and o.o.d. evaluation $\tilde{p}(D)$ (bottom). For visualization, the adjacency matrices **(a)** are topologically sorted, the causal mechanisms **(b)** have two parents, where $c$ and $\ell$ are output and length scales of the underlying GP, and the noise **(c)** is shown as a function of one parent, where dashed lines indicate 0.66, 0.95, and 0.999 coverage.

from a Gaussian process with squared exponential kernel (Rahimi and Recht, 2007). In the out-of-distribution (o.o.d.) setting $\tilde{p}(D)$, we sample the linear function and kernel parameters from the tails of $p(D)$ and unseen value ranges. Moreover, we simulate homoscedastic Gaussian noise in the training distribution $p(D)$ but test on heteroscedastic Cauchy and Laplacian noise o.o.d. that is induced by randomly drawn, nonlinear functions $h_j$. In LINEAR and RFF, interventions set variables to random values and are performed on a subset of target variables containing half of the nodes.

In addition to SCMs, we consider the challenging domain of GRNs (GRN) using the simulator of Dibaeinia and Sinha (2020). Contrary to SCMs, gene expression samples correspond to draws from the steady state of a stochastic dynamical system that varies between cell types (Huynh-Thu and Sanguinetti, 2019). In the o.o.d. setting, the parameters sampled for the GRN simulator are drawn from significantly wider ranges. In addition, we use the noise levels of different single-cell RNA sequencing technologies, which were calibrated on real datasets. In GRN, interventions are performed on all nodes and correspond to gene knockouts, forcing the transcription rate of a variable to zero.

**Causal structures** $p(G)$ Following prior work, we use random graph models and known biological networks to sample ground-truth causal structures. In all three domains, the training data distribution $p(D)$ is induced by simple Erdős-Rényi and scale-free graphs (Erdős and Rényi, 1959; Barabási and Albert, 1999). In the o.o.d. setting, $\tilde{p}(D)$ of the LINEAR and RFF domains are simulated using causal structures from the Watts-Strogatz model, capturing small-world phenomena (Watts and Strogatz, 1998); the stochastic block model, generalizing Erdős-Rényi to community structures (Holland et al., 1983); and geometric random graphs, modeling connectivity based on spatial distance (Gilbert, 1961). In the GRN domain, we use subgraphs of the known *S. cerevisiae* and *E. coli* GRNs and their effect signs whenever known. To extract these subgraphs, we use the procedure by Marbach et al. (2009) to maintain structural patterns like motifs and modularity (Ravasz et al., 2002; Shen-Orr et al., 2002).

To illustrate the distribution shift from $p(D)$ to $\tilde{p}(D)$, Figure 2 shows a set of graph, mechanism, and noise distribution samples in the RFF domain. In Appendix A, we give the detailed parameter configurations and functions defining $p(D)$ and $\tilde{p}(D)$ in the three domains. We also provide details on the simulator by Dibaeinia and Sinha (2020) and subgraph extraction (Marbach et al., 2009).

## 5.2 Evaluation Metrics

All experiments throughout this paper are conducted on datasets that AVICI has never seen during training, regardless of whether we evaluate the predictive performance in-distribution or o.o.d. To assess how well a predicted structure reflects the ground truth, we report the structural Hamming distance (SHD) and the structural intervention distance (SID) (Peters and Bühlmann, 2015). While the SHD simply reflects the graph edit distance, the SID quantifies the closeness of two graphs in terms of their interventional adjustment sets. For these metrics and for single-edge precision, recall, and F1 score, we convert the posterior probabilities predicted by AVICI to hard predictions using a threshold of $0.5$. We evaluate the uncertainty estimates by computing the areas under the precision-recall curve (AUPRC) and the receiver operating characteristic (AUROC) (Friedman and Koller, 2003). How

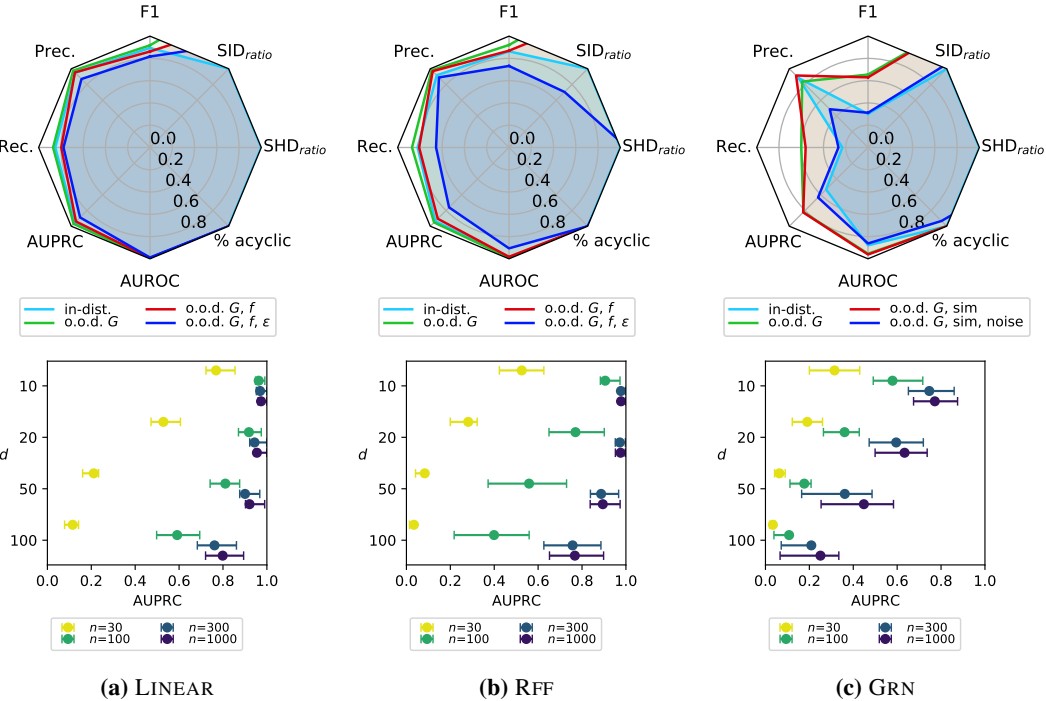

**(a)** LINEAR       **(b)** RFF       **(c)** GRN

**Figure 3: Generalization properties of the inference model $f_\phi$.** Top row plots show performance metrics of AVICI under increasing distributional shift given $n = 1000$ observations for $d = 30$ variables. $\text{SID}_{\text{ratio}}$ is defined as $\text{SID}_{\text{in-dist.}}/\text{SID}$ and analogously for SHD. Thus, higher is better for all metrics. Bottom row shows the in-distribution AUPRC for various $d$ as we vary the number of observations provided to AVICI. The datasets contain interventional data (cf. Section 5.1). All values are the mean over fifteen random task instances. Error bars indicate the interquartile range.

well these uncertainty estimates are calibrated is quantified with the expected calibration error (ECE) (DeGroot and Fienberg, 1983). More details on the metrics are given in Appendix B.

## 5.3 Inference Model Configuration

We train three inference models overall, one for each domain, and perform all experiments on these three trained models, both when predicting from only observational and from interventional data. During training, the datasets sampled from $p(D)$ have $d = 2$ to $50$ variables and $n = 200$ samples. With probability 0.5, these training datasets contain 50 interventional samples. The inference models in the three domains share identical hyperparameters for the architecture and optimization, except for the dropout rate. We add the acyclicity constraint for the SCM domains LINEAR and RFF. Details on the optimization and architecture are given in Appendix C.

## 6 Experimental Results

### 6.1 Out-Of-Distribution Generalization

**Sensitivity to distribution shift** In our first set of experiments, we study the generalization capabilities of our inference models across the spectrum of test distributions described in Section 5.1. We perform causal discovery from $n = 1000$ observations in systems of $d = 30$ variables. Starting from the training distribution $p(D)$, we incrementally introduce the described distribution shifts in the causal structures, causal mechanisms, and finally noise, where fully o.o.d. corresponds to $\tilde{p}(D)$. The top row of Figure 3 visualizes the results of an empirical sensitivity analysis. The radar plots disentangle how combinations of the three o.o.d. aspects, i.e., graphs, mechanisms, and noise, affect the empirical performance in the three domains LINEAR, RFF, and GRN. In addition to the metrics in Section 5.2, we also report the percentage of predicted graphs that are acyclic.

In the LINEAR domain, AVICI performs very well in all metrics and hardly suffers under distribution shift. In contrast, GRN is the most challenging problem domain and the performance degrades

**Table 1: Benchmarking results ($d = 30$ variables).** Mean SID ($\downarrow$) and F1 score ($\uparrow$) with standard error of all methods on 30 random task instances. Methods in the top section use only observational data, in the bottom section both observational and interventional data. We highlight the best result of each section and those within its 95% confidence interval according to an unequal variances $t$-test.

| Algorithm | LINEAR | | RFF | | GRN | |
|---|---|---|---|---|---|---|
| | SID | F1 | SID | F1 | SID | F1 |
| **GES** | **215.6** (35.0) | 0.548 (0.03) | **346.3** (44.4) | 0.285 (0.03) | **573.6** (29.2) | **0.058** (0.01) |
| **LiNGAM** | 413.4 (48.4) | 0.369 (0.04) | 410.3 (47.6) | 0.238 (0.02) | 617.5 (31.7) | **0.044** (0.01) |
| **PC** | 400.5 (53.7) | 0.338 (0.03) | **370.1** (51.2) | 0.421 (0.03) | 594.0 (30.0) | **0.061** (0.01) |
| **DAG-GNN** | 474.5 (50.8) | 0.154 (0.01) | 425.3 (50.2) | 0.221 (0.03) | 588.7 (36.6) | **0.078** (0.02) |
| **GraN-DAG** | 466.0 (54.3) | 0.200 (0.03) | 328.6 (48.4) | **0.476** (0.05) | 582.4 (33.4) | **0.073** (0.02) |
| **AVICI** (ours) | **145.6** (21.5) | **0.672** (0.04) | **255.1** (48.2) | **0.618** (0.06) | 641.7 (34.7) | 0.000 (0.00) |
| **GIES** | **120.8** (26.2) | 0.736 (0.03) | 304.8 (44.0) | 0.338 (0.04) | 545.5 (26.9) | 0.092 (0.01) |
| **IGSP** | 244.0 (34.4) | 0.559 (0.02) | 374.1 (45.0) | 0.407 (0.04) | 597.4 (31.7) | 0.057 (0.01) |
| **DCDI** | 383.5 (45.1) | 0.327 (0.03) | **282.8** (46.3) | 0.409 (0.04) | 590.9 (30.6) | 0.075 (0.02) |
| **AVICI** (ours) | **110.9** (19.3) | **0.819** (0.02) | **192.7** (44.8) | **0.707** (0.06) | **416.9** (47.1) | **0.338** (0.06) |

more significantly for the o.o.d. scenarios. We observe that AVICI can perform better under certain distribution shifts than in-distribution, e.g., in GRN. This is because AVICI empirically performs better at predicting edges adjacent to large-degree nodes, a common feature of the *E. coli* and *S. cerevisiae* graphs not present in the Erdős-Rényi training structures. We also find that acyclicity is perfectly satisfied for LINEAR and RFF and that AUPRC and AUROC do not suffer as much from distributional shift as the metrics based on thresholded point estimates.

In Appendix E.1, we additionally report results for generalization from LINEAR to RFF and vice versa, i.e., to entirely unseen function classes of causal mechanisms in addition to the previous o.o.d. shifts.

**Generalization to unseen problem sizes**    In addition to the sensitivity to distribution shift, we study the ability to generalize to unseen problem sizes. The bottom row of Figure 3 illustrates the AUPRC for the edge predictions of AVICI when varying $d$ and $n$ on unseen in-distribution data. The predictions improve with the number of data points $n$ while exhibiting diminishing marginal improvement when seeing additional data. Moreover, the performance decreases smoothly as the number of variables $d$ increases and the task becomes harder. Most importantly, this robust behavior can be observed well beyond the settings used during training ($n = 200$ and $d \leq 50$).

## 6.2    Benchmarking

Next, we benchmark AVICI against existing algorithms. Using only observational data, we compare with the PC algorithm (Spirtes et al., 2000), GES (Chickering, 2003), LiNGAM (Shimizu et al., 2006), DAG-GNN (Yu et al., 2019), and GraN-DAG (Lachapelle et al., 2020). Mixed with interventional data, we compare with GIES (Hauser and Bühlmann, 2012), IGSP (Wang et al., 2017), and DCDI (Brouillard et al., 2020). We tune the important hyperparameters of each baseline on held-out task instances of each domain. When computing the evaluation metrics, we favor methods that only predict (interventional) Markov equivalence classes by orienting undirected edges correctly when present in the ground truth. Details on the baselines are given in Appendix D.

The benchmarking is performed on the fully o.o.d. domain distributions $\tilde{p}(D)$, i.e., under distribution shifts on causal graphs, mechanisms, and noise distributions w.r.t. the training distribution of AVICI. Table 1 shows the SID and F1 scores of all methods given $n = 1000$ observations for $d = 30$ variables. We find that the AVICI model trained on LINEAR outperforms all baselines, both given observational or interventional data, despite operating under significant distribution shift. Only GIES achieves comparable accuracy. The same holds for RFF, where GraN-DAG and DCDI perform well but ultimately do not reach the accuracy of AVICI.

In the GRN domain, where inductive biases are most difficult to specify, classical methods fail to infer plausible graphs. However, provided interventional data, AVICI can use its learned inductive bias to infer plausible causal structures from the noisy gene expressions, even under distribution shift. This is a promising step towards reliable structure discovery in fields like molecular biology. Even without gene knockout data, AVICI achieves nontrival AUROC and AUPRC while classical methods predict close to randomly (Table 9 in Appendix E; see also Dibaeinia and Sinha 2020; Chen and Mar 2018).

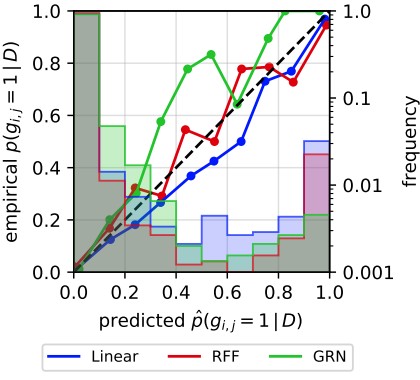

| | LINEAR | RFF | GRN |
|---|---|---|---|
| **GES**[*] | 0.031 (0.00) | 0.068 (0.02) | 0.092 (0.01) |
| **LiNGAM**[*] | 0.066 (0.02) | 0.054 (0.01) | 0.053 (0.01) |
| **PC**[*] | 0.036 (0.00) | **0.033** (0.01) | 0.065 (0.01) |
| **DAG-GNN**[*] | 0.078 (0.01) | 0.063 (0.01) | 0.063 (0.01) |
| **GraN-DAG**[*] | 0.046 (0.01) | **0.042** (0.01) | 0.199 (0.05) |
| **AVICI** (ours) | **0.013** (0.00) | **0.024** (0.01) | **0.018** (0.00) |
| **GIES**[*] | 0.027 (0.00) | 0.074 (0.02) | 0.094 (0.01) |
| **IGSP**[*] | 0.042 (0.01) | 0.083 (0.01) | 0.077 (0.01) |
| **DCDI**[*] | 0.068 (0.01) | 0.087 (0.02) | 0.170 (0.03) |
| **DiBS** | 0.056 (0.02) | **0.035** (0.01) | 0.093 (0.01) |
| **AVICI** (ours) | **0.011** (0.00) | **0.022** (0.01) | **0.024** (0.01) |

[*] Nonparametric DAG bootstrap (Friedman et al., 1999)

(a)            (b)

**Figure 4: Uncertainty calibration ($d = 30$ variables).** As previously, datasets are held-out and o.o.d. in terms of graph, parameters, and noise. **(a)** Calibration plots for AVICI aggregating the predictions for ten test cases of each domain. The histograms on the right $y$-axis show the frequency of predictions at each confidence level. **(b)** ECE ($\downarrow$) with standard error averaged over ten test cases. Methods in the top (bottom) section use observational (and interventional) data. We highlight the best result and those within its $95\%$ confidence interval according to an unequal variances $t$-test.

Results for in-distribution data and for larger graphs of $d = 100$ variables are given in Appendices E.2 and E.3. In Appendix E.4, we also report results for a real proteomics dataset (Sachs et al., 2005).

**Uncertainty quantification** Using metrics of calibration, we can evaluate the degree to which predicted edge probabilities are consistent with empirical edge frequencies (DeGroot and Fienberg, 1983; Guo et al., 2017). We say that a predicted probability $p$ is calibrated if we empirically observe an event in $(p \cdot 100)\%$ of the cases. When plotting the observed edge frequencies against their predicted probabilities, a calibrated algorithm induces a diagonal line. The expected calibration error (ECE) represents the weighted average deviation from this diagonal. For further details, see Appendix B.

Since the baseline algorithms only infer point estimates of the causal structure, we use the non-parametric DAG bootstrap to estimate edge probabilities (Friedman et al. 1999, Appendix D). We additionally compare AVICI with DiBS, which infers Bayesian posterior edge probabilities like AVICI (Lorch et al., 2021). Figure 4 gives the calibration plots for AVICI and Table 4b the ECE for all methods. In each domain, the marginal edge probabilities predicted by AVICI are the most calibrated in terms of ECE. Moreover, Figure 4a shows that AVICI closely traces the perfect calibration line, which highlights its accurate uncertainty calibration across the probability spectrum.

In Appendix E.5, we additionally report AUROC and AUPRC metrics for all methods. We also provide calibration plots analogous to Figure 4 for the baselines (Figure 6), which often show vastly overconfident predictions where the calibration line is far below the diagonal.

### 6.3 Ablations

Finally, we analyze the importance of key architecture components of the inference network $f_\phi$. Focusing on the RFF domain, we train several additional models and ablate single architecture components. We vary the network depth $L$, the axes of attention, the representation of $\theta$, and the number of training steps for $\phi$. All other aspects of the model, training and data simulation remain unchanged.

Table 2 summarizes the results. Most noticeably, we find that the performance drops significantly when attending only over axis $d$ and aggregating information over axis $n$ only once through pooling after the $2L$ self-attention layers. Attending only over axis $n$ is not sensible since variable interactions are not processed until the prediction of $\theta$, but we still include the results for completeness.

We also test an alternative variational parameter model given by $\theta_{i,j} = \phi_\theta^\top \tanh\left(\phi_u^\top \mathbf{u}^i + \phi_v^\top \mathbf{v}^j\right)$ that uses an additional, learned vector $\phi_\theta$ and matrices $\phi_u, \phi_v$. This model has been used in related causal discovery work for searching over high-scoring causal DAGs (Zhu et al., 2020) and is a relational network (Santoro et al., 2017). This variant also satisfies permutation equivariance (cf. Section 4.2) since it applies the same MLP elementwise to each edge pair $[\mathbf{u}^i, \mathbf{v}^j]$. Ultimately, we find no statistically significant difference in performance to our simpler model in Eq. (6), hence we opt for less parameters and a lower memory requirement.

**Table 2: Ablations of the architecture of $f_\phi$.** Models are evaluated on 100 interventional datasets of $d = 30$ variables in the RFF domain. Top row ($\star$) corresponds to the model used in the main experiments. In (a), we vary the number of blocks $L$; in (b), the axes over which attention is performed; in (c), the generative model of the variational parameters; in (d), the number of update steps of $\phi$. We again highlight the best result and those within its $95\%$ $t$-test confidence interval.

| | $L$ | ax. $d$ | ax. $n$ | $\theta$ model | steps | RFF (in-dist.) SID | RFF (in-dist.) AUPRC | RFF (o.o.d.) SID | RFF (o.o.d.) AUPRC |
|---|---|---|---|---|---|---|---|---|---|
| ($\star$) | 8 | ✓ | ✓ | Eq. (6) | 300k | **65.2** (8.4) | **0.972** (0.00) | **221.5** (24.7) | **0.650** (0.03) |
| (a) | 1 | | | | | 267.2 (22.0) | 0.635 (0.01) | 394.2 (28.4) | 0.242 (0.02) |
| | 2 | | | | | 195.9 (18.5) | 0.825 (0.01) | 343.1 (27.1) | 0.400 (0.03) |
| | 4 | | | | | 116.6 (13.1) | 0.937 (0.01) | **264.0** (24.8) | 0.566 (0.03) |
| (b) | | ✓ | | | | 351.5 (27.9) | 0.552 (0.01) | 414.2 (29.5) | 0.209 (0.02) |
| | | | ✓ | | | 416.8 (29.6) | 0.256 (0.01) | 390.2 (27.6) | 0.078 (0.01) |
| (c) | | | | (Santoro et al., 2017) | | **72.4** (9.2) | **0.971** (0.00) | **225.7** (25.2) | **0.634** (0.03) |
| (d) | | | | | 100k | 96.9 (11.6) | 0.955 (0.00) | **259.3** (26.6) | **0.589** (0.04) |

Lastly, Table 2 shows that the causal discovery performance of AVICI scales up monotonically with respect to network depth and training time. Even substantially smaller models of $L = 4$ or shorter training times achieve an accuracy that is on par with most baselines (cf. Table 1). Our main models ($\star$) have a moderate size of $4.2 \times 10^6$ parameters, which amounts to only 17.0 MB at f32 precision. Performing causal discovery (computing a forward pass) given on a trained model takes only a few seconds on CPU.

## 7  Discussion

We proposed AVICI, a method for inferring causal structure by performing amortized variational inference over an arbitrary data-generating distribution. Our approach leverages the insight that inductive biases crucial for statistical efficiency in structure learning might be more easily encoded in a simulator than in an inference technique. This is reflected in our experiments, where AVICI solves structure learning problems in complex domains intractable for existing approaches (Dibaeinia and Sinha, 2020). Our method can likely be extended to other typically difficult domains, including settings where we cannot assume causal sufficiency (Bhattacharya et al., 2021). Our approach will continually benefit from ongoing efforts in developing (conditional) generative models and domain simulators.

Using AVICI still comes with several trade-offs. First, while optimizing the dual program empirically induces acyclicity, this constraint is not satisfied with certainty using the variational family considered here. Moreover, similar to most amortization techniques (Amos, 2022), AVICI gives no theoretical guarantees of performance. Some classical methods can do so in the infinite sample limit given specific assumptions on the data-generating process (Peters et al., 2017). However, future work might obtain guarantees for AVICI that are similar to learning theory results for the bivariate causal discovery case (Lopez-Paz et al., 2015).

Our experiments demonstrate that our inference models are highly robust to distributional shift, suggesting that the trained models could be useful out-of-the-box in causal structure learning tasks outside the domains studied in this paper. In this context, fine-tuning a pretrained AVICI model on labeled real-world datasets is a promising avenue for future work. To facilitate this, our code and models are publicly available at: `https://github.com/larslorch/avici`.

**Acknowledgments and Disclosure of Funding**

We thank Alexander Neitz, Giambattista Parascandolo, and Frederik Träuble for their feedback and the reviewers for their helpful comments. This research was supported by the European Research Council (ERC) under the European Union's Horizon 2020 research and innovation program grant agreement no. 815943 and the Swiss National Science Foundation under NCCR Automation, grant agreement 51NF40 180545. Jonas Rothfuss was supported by an Apple Scholars in AI/ML fellowship.

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
