# A    Domain Specification and Simulation

In this section, we define the training and test distributions $p(D)$ and $\tilde{p}(D)$ concretely in terms of the parameters and notation introduced in the main text. Based on these definitions, Table 3 summarizes all parameters of the data-generating processes for LINEAR and RFF and specifies how they are sampled for a random task instance. Table 4 lists the same specifications for the GRN domain. The notation and parameters are defined in the following subsections.

**Table 3:** Specification of the training and out-of-distribution data-generating processes $p(D)$ and $\tilde{p}(D)$ for the LINEAR and RFF domain. All specifications except the mechanism function type are the same for the two domains. For each random task instance, the parameter configurations are sampled uniformly randomly from all possible combinations of the sets of options. The graph model classes are sampled in equal proportions out-of-distribution. Empty fields indicate that the component is not part of the distribution.

| | IN-DISTRIBUTION $p(D)$ | | OUT-OF-DISTRIBUTION $\tilde{p}(D)$ | |
|---|---|---|---|---|
| **Graph** | | | | |
| Erdős-Rényi | expected edges/node | $\in \{1, 2, 3\}$ | | |
| Scale-free (in-degree) | edges/node | $\in \{1, 2, 3\}$ | | |
| | attach. power $\alpha$ | $\in \{1.0\}$ | | |
| Scale-free (out-degree) | edges/node | $\in \{1, 2, 3\}$ | edges/node | $\in \{2\}$ |
| | attach. power $\alpha$ | $\in \{1.0\}$ | attach. power $\alpha$ | $\in \{0.5, 1.5\}$ |
| Watts-Strogatz | | | lattice dim. $k$ | $\in \{2, 3\}$ |
| | | | rewire prob. | $\in \{0.3\}$ |
| Stochastic Block Model | | | expected edges/node | $\in \{2\}$ |
| | | | blocks | $\in \{5, 10\}$ |
| | | | damp. inter-block prob. | $\in \{0.1\}$ |
| Geometric Random Graphs | | | radius | $\in \{0.1\}$ |
| | | | | |
| **Mechanism** | | | | |
| Linear function[a] | weights $\mathbf{w}$ | $\sim \mathrm{Unif}_\pm(1, 3)$ | weights $\mathbf{w}$ | $\sim \mathrm{Unif}_\pm(0.5, 2)$ |
| | | | | $\sim \mathrm{Unif}_\pm(2, 4)$ |
| | bias $b$ | $\sim \mathrm{Unif}(-3, 3)$ | bias $b$ | $\sim \mathrm{Unif}(-3, 3)$ |
| Random Fourier function[b] | SE length scale $\ell$ | $\sim \mathrm{Unif}(7, 10)$ | SE length scale $\ell$ | $\sim \mathrm{Unif}(5, 8)$ |
| | | | | $\sim \mathrm{Unif}(8, 12)$ |
| | SE output scale $c$ | $\sim \mathrm{Unif}(10, 20)$ | SE output scale $c$ | $\sim \mathrm{Unif}(8, 12)$ |
| | | | | $\sim \mathrm{Unif}(18, 22)$ |
| | bias $b$ | $\sim \mathrm{Unif}(-3, 3)$ | bias $b$ | $\sim \mathrm{Unif}(-3, 3)$ |
| | | | | |
| **Noise** (indiv. per variable) | | | | |
| $\mathcal{N}(0, \sigma^2)$ | $\sigma$ | $\sim \mathrm{Unif}(0.2, 2)$ | | |
| Laplace$(0, \sigma^2)$ | | | $\sigma^2(\mathbf{x}_{\mathrm{pa}_j})$ (heterosced.) | $\sim p(h_{\mathrm{rff}})$ |
| Cauchy$(0, \sigma^2)$ | | | $\sigma^2(\mathbf{x}_{\mathrm{pa}_j})$ (heterosced.) | $\sim p(h_{\mathrm{rff}})$ |
| | | | | |
| **Interventions** | | | | |
| Target nodes | random 50% of nodes | | random 50% of nodes | |
| Intervention values | $x_j$ | $\sim \mathrm{Unif}_\pm(1, 3)$ | $x_j$ | $\sim \mathrm{Unif}_\pm(1, 5)$ |

[a] Only LINEAR domain
[b] Only RFF domain

Aliases:
• $\mathrm{Unif}_\pm(a, b)$: uniform mixture of $\mathrm{Unif}(a, b)$ and $\mathrm{Unif}(-b, -a)$
• $p(h_{\mathrm{rff}})$: distribution over heteroscedastic noise scale functions, induced by the squash function $h_{\mathrm{rff}}(\mathbf{x}) = \log(1 + \exp(g_{\mathrm{rff}}(\mathbf{x}))$
  and random Fourier feature functions $g_{\mathrm{rff}}(\mathbf{x})$ with SE length scale $\ell = 10$ and output scale $c = 2$ (cf. RFF domain)

**Table 4:** Specification of the training and out-of-distribution data-generating processes $p(D)$ and $\tilde{p}(D)$ for the GENE domain. For each random task instance, the parameter configurations are sampled uniformly randomly from all possible combinations of the sets of options. The graph model classes are sampled in equal proportions out-of-distribution. Empty fields indicate that the component is not part of the distribution.

| | IN-DISTRIBUTION $p(D)$ | | OUT-OF-DISTRIBUTION $\tilde{p}(D)$ | |
|---|---|---|---|---|
| **Graph** | | | | |
| Erdős-Rényi | expected edges/node | $\in \{1,2,3\}$ | | |
| Scale-free (out-degree) | edges/node | $\in \{1,2,3\}$ | | |
| | attach. power $\alpha$ | $\in \{0.5, 0.8, 1.0, 1.2, 1.5\}$ | | |
| *E. coli* subgraph (Marbach et al., 2009) | | | top-$p$ perc. modular | $\in \{0.2\}$ |
| *S. cerevisiae* subgraph (Marbach et al., 2009) | | | top-$p$ perc. modular | $\in \{0.2\}$ |
| **Mechanism** | | | | |
| GRN simulator (Dibaeinia and Sinha, 2020) | no. cell types | $\in \{5\}$ | no. cell types | $\in \{10\}$ |
| | decay rates $\lambda$ | $\in \{0.7, 0.8, 0.9\}$ | decay rates $\lambda$ | $\in \{0.5, 1.5\}$ |
| | system noise scale $\xi$ | $\in \{0.9, 1.0, 1.1\}$ | system noise scale $\xi$ | $\in \{0.5, 1.5\}$ |
| | Hill function coeff. $\gamma$ | $\in \{1.9, 2.0, 2.1\}$ | Hill function coeff. $\gamma$ | $\in \{1.5, 2.5\}$ |
| | MR prod. rate $b$ | $\sim \text{Unif}(1,3)$ | MR prod. rate $b$ | $\sim \text{Unif}(0.5, 2)$ |
| | | | | $\sim \text{Unif}(2,4)$ |
| | interactions $k$ | $\sim \text{Unif}(1,5)$ | interactions $k$ | $\sim \text{Unif}(1,3)$ |
| | | | | $\sim \text{Unif}(3,7)$ |
| | signs($k$) per node | $\sim \text{Bern}(p)$ where $p$ | signs($k$) per node | from *E. coli* |
| | | $\sim \text{Beta}(1,1)$ | | or $\sim \text{Bern}(p)$ |
| | | $\sim \text{Beta}(0.5, 0.5)$ | | (cf. Sec. A.1.2) |
| **Measurement Noise** | | | | |
| Platform[†] | 10X chromium | $p_{\text{outlier}} \in \{0.01\}$ | | |
| | | $\mu_{\text{outlier}} \in \{3.0, 5.0\}$ | | |
| | | $\sigma_{\text{outlier}} \in \{1.0\}$ | | |
| | | $\mu_{\text{lib}} \in \{4.5, 6.0\}$ | | |
| | | $\sigma_{\text{lib}} \in \{0.3, 0.4, 0.7\}$ | | |
| | | $\delta \in \{45, 74, 82\}$ | | |
| | | $\tau \in \{8.0\}$ | | |
| | | | Illumina HiSeq2000 | $p_{\text{outlier}} \in \{0.01\}$ |
| | | | | $\mu_{\text{outlier}} \in \{0.8\}$ |
| | | | | $\sigma_{\text{outlier}} \in \{1.0\}$ |
| | | | | $\mu_{\text{lib}} \in \{7.0\}$ |
| | | | | $\sigma_{\text{lib}} \in \{0.4\}$ |
| | | | | $\delta \in \{80\}$ |
| | | | | $\tau \in \{8.0\}$ |
| | | | Drop-seq | $p_{\text{outlier}} \in \{0.01\}$ |
| | | | | $\mu_{\text{outlier}} \in \{3.0\}$ |
| | | | | $\sigma_{\text{outlier}} \in \{1.0\}$ |
| | | | | $\mu_{\text{lib}} \in \{4.4\}$ |
| | | | | $\sigma_{\text{lib}} \in \{0.8\}$ |
| | | | | $\delta \in \{85\}$ |
| | | | | $\tau \in \{8.0\}$ |
| | | | Smart-seq | $p_{\text{outlier}} \in \{0.01\}$ |
| | | | | $\mu_{\text{outlier}} \in \{4.5\}$ |
| | | | | $\sigma_{\text{outlier}} \in \{1.0\}$ |
| | | | | $\mu_{\text{lib}} \in \{10.8\}$ |
| | | | | $\sigma_{\text{lib}} \in \{0.55\}$ |
| | | | | $\delta \in \{92\}$ |
| | | | | $\tau \in \{2.0\}$ |
| **Interventions** | | | | |
| Target nodes | all nodes | | all nodes | |
| Intervention type | gene knockout | | gene knockout | |

[†] Noise specifications were collected from calibrations performed by Dibaeinia and Sinha (2020) on real datasets generated by the different scRNA-seq platforms.

## A.1    Causal Structures

### A.1.1    Random graph models

In Erdős-Rényi graphs, each edge is sampled independently with a fixed probability (Erdős and Rényi, 1959). We scale this probability to obtain $O(d)$ edges in expectation. Scale-free graphs are generated by a sequential preferential attachment process, where in- or outgoing edges of node $i$ to the previous $i - 1$ nodes are sampled with probability $\propto \deg(j)^\alpha$ (Barabási and Albert, 1999). Watts-Strogatz graphs are $k$-dimensional lattices, whose edges get rewired globally to random nodes with a specified probability (Watts and Strogatz, 1998). The stochastic block model generalizes Erdős-Rényi to capture community structure. Splitting the nodes into a random partition of so-called blocks, the inter-block edge probability is dampened by a multiplying factor compared to the intra-block probability, also tuned to result in $O(d)$ edges in expectation (Holland et al., 1983). Lastly, geometric random graphs model connectivity based on two-dimensional Euclidian distance within some radius, where nodes are randomly placed inside the unit square (Gilbert, 1961).

For undirected random graph models, we orient edges by selecting the upper-triangular half of the adjacency matrix. The classes of random graph models are sampled in equal proportion when generating a set of evaluation datasets (Tables 3 and 4).

### A.1.2    Subgraph Extraction from Real-World Networks

For the evaluation in the GRN domain, we sample realistic causal graphs by extracting subgraphs from the known *E. coli* and *S. cerevisiae* regulatory networks. For this, we rely on the procedure by Marbach et al. (2009), which is also used by Schaffter et al. (2011) and Dibaeinia and Sinha (2020). Their graph extraction method is carefully designed to capture the structural properties of biological networks by preserving the functional and structural properties of the source network.

**Algorithm**    The procedure extracts a random subgraph of the source network by selecting a subset of nodes $\mathcal{V}$, and then returning the graph containing all edges from the source network covered by $\mathcal{V}$. Starting from a random seed node, the algorithm proceeds by iteratively adding new nodes to $\mathcal{V}$. In each step, this new node is selected from the set of neighbors of the current set $\mathcal{V}$. The neighbor to be added is selected greedily such that the resulting subgraph has maximum modularity (Marbach et al., 2009).

To introduce additional randomness, Marbach et al. (2009) propose to randomly draw the new node from the set of neighbors inducing the top-$p$ percent of the most modular graphs. In our experiments, we adopt the latter with $p = 20$ percent, similar to Schaffter et al. (2011). The original method of Marbach et al. (2009) is intended for undirected graphs. Thus, we use the undirected skeleton of the source network for the required modularity and neighborhood computation.

**Real GRNs**    We take the *E. coli* and *S. cerevisiae* regulatory networks as provided by the GeneNetWeaver repository,[1] which have 1565 and 4441 nodes (genes), respectively. For *E. coli*, we also know the true signs of a large proportion of causal effects. When extracting a random subgraph from *E. coli*, we take the true signs of the effects and map them onto the randomly sampled interaction terms $k \in \mathbb{R}^{d \times d}$ used by SERGIO; cf. Section A.2.2. When the interaction signs are unknown or uncertain in *E. coli*, we impute a random sign in the interaction terms $k$ of SERGIO based on the frequency of known positive and negative signs in the *E. coli* graph.

Empirically, individual genes in *E. coli* tend to predominantly have either up- or down-regulating effects on their causal children. To capture this aspect in *S. cerevisiae* also, we fit the probability of an up-regulating effect caused by a given gene in *E. coli* to a Beta distribution. For each node $j$ in an extracted subgraph of *S. cerevisiae*, we draw a probability $p_j$ from this Beta distribution and then sample the effect signs for the outgoing edges of node $j$ using $p_j$. As a result, the genes in the subgraphs of *S. cerevisiae* individually also have mostly up- or down-regulating effects. Maximum likelihood estimation for this Beta distribution yielded $\alpha = 0.2588$ and $\beta = 0.2499$.

The *E. coli* and *S. cerevisiae* graphs and effect signs used in the experiments are taken from the GeneNetWeaver repository (Schaffter et al., 2011) (MIT License).

---

[1] https://github.com/tschaffter/genenetweaver

## A.2 Data-Generating Processes

### A.2.1 Structural Causal Models

In the LINEAR and RFF domains, the data-generating processes are modeled by structural causal models (SCMs). In this work, we consider SCMs with causal mechanisms that model each causal variable $x_j$ given its parents $\mathbf{x}_{\text{pa}(j)}$ as

$$x_j \leftarrow f_j(\mathbf{x}_{\text{pa}(j)}, \epsilon_j) = f_j(\mathbf{x}_{\text{pa}(j)}) + h_j(\mathbf{x}_{\text{pa}(j)})\epsilon_j \tag{10}$$

where the noise $\epsilon_j$ is additive and may be heteroscedastic through an input-dependent noise scale $h_j(\mathbf{x}_{\text{pa}(j)})$. Even in the homogeneous noise setting, the scale of each noise distribution $p(\epsilon_j)$ is random and thus different for each variable $x_j$. We write $\mathbf{x}_{\text{pa}(j)}$ when indexing $\mathbf{x}$ at the parents of node $j$. In the heteroscedastic setting, we parameterize the noise scales as $h_j(\mathbf{x}) = \log(1+\exp(g_j(\mathbf{x})))$ for a set of nonlinear functions $g_j$.

Prior to performing inference with AVICI or any baseline, each set of SCM observations $D$ is standardized variable-wise by subtracting its mean and dividing by its standard deviation, so that each $x_j$ has mean 0 and variance 1, avoiding potential varsortability bias (Reisach et al., 2021).

In the LINEAR domain, the functions $f_j$ are given by affine transforms

$$f_j(\mathbf{x}_{\text{pa}(j)}) = \mathbf{w}_j^\top \mathbf{x}_{\text{pa}(j)} + b_j \tag{11}$$

whose weights $\mathbf{w}_j$ and bias $b_j$ are sampled independently for each $f_j$. In the RFF domain, the functions $f_j$ modeling each causal variable $x_j$ given its parents $\mathbf{x}_{\text{pa}(j)}$ are drawn from a Gaussian Process

$$f_j \sim \mathcal{GP}(b_j, k_j) \tag{12}$$

with bias $b_j$ and squared exponential (SE) kernel $k_j(\mathbf{x}, \mathbf{x}') = c_j^2 \exp\left(-\|\mathbf{x} - \mathbf{x}'\|_2^2/2\ell_j^2\right)$ with length scale $\ell_j$ and output scale $c_j$. The parameters $b_j$, $c_j$, and $\ell_j$ are sampled independently for each variable $j$. To obtain explicit function draws $f_j$ from the GP, we approximate $f_j$ with random Fourier features (Rahimi and Recht, 2007). Specifically, we can obtain $f_j \sim \mathcal{GP}(b_j, k(\mathbf{x}, \mathbf{x}'))$ for a SE kernel $k$ with length scale $\ell_j$ and output scale $c_j$ by sampling

$$f_j(\mathbf{x}_{\text{pa}(j)}) = b_j + c_j \sqrt{\frac{2}{M}} \sum_{m=1}^{M} \alpha^{(m)} \cos\left(\frac{1}{\ell_j} \boldsymbol{\omega}^{(m)} \cdot \mathbf{x}_{\text{pa}(j)} + \delta^{(m)}\right) \tag{13}$$

with $\alpha^{(m)} \sim \mathcal{N}(0,1)$, $\boldsymbol{\omega}^{(m)} \sim \mathcal{N}(0,\mathbf{I})$, and $\delta^{(m)} \sim \text{Unif}(0, 2\pi)$. Throughout this work, we use $M = 100$. The function draws become faithful GP samples as $M \to \infty$ (Rahimi and Recht, 2007). When $x_j$ is a root node and thus has no parents, $f_j$ is a constant.

### A.2.2 Single-Cell Gene Expression Data

In the GRN domain, our goal is to evaluate causal discovery from realistic gene expression data. There exist several models to simulate the mechanisms, intervention types, and technical measurement noise underlying single-cell expression data of gene regulatory networks (Schaffter et al., 2011; Huynh-Thu and Sanguinetti, 2019; Dibaeinia and Sinha, 2020). We use the simulator by Dibaeinia and Sinha (2020) (SERGIO) because it resembles the data collected by modern high-throughput single-cell RNA sequencing (scRNA-seq) technologies. Related genomics simulators, for example, GeneNetWeaver (Schaffter et al., 2011), were developed for the simulation of microarray gene expression platforms. In the following, we give an overview of how to simulate scRNA-seq data with SERGIO. Dibaeinia and Sinha (2020) provide all the details and additional background from the related literature.

**Simulation** Given a causal graph over $d$ genes and a specification of the simulation parameters, SERGIO generates a synthetic scRNA-seq dataset $D$ in two stages. The $n$ observations in $D$ correspond to $n$ cell samples, that is, the expressions of the $d$ genes recorded in a single cell corresponds to one row in $D$.

In the first stage, SERGIO simulates clean gene expressions by sampling randomly-timed snapshots from the steady state of a dynamical system. In this regulatory process, the genes are expressed at rates influenced by other genes using the chemical Langevin equation, similar to Schaffter et al. (2011) and

(Dibaeinia and Sinha, 2020). The source nodes in the causal graph $G$ are denoted master regulators (MRs), whose expressions evolve at constant production and decay rates. The expressions of all downstream genes evolve nonlinearly under production rates caused by the expression of their causal parents in $G$. Cell *types* are defined by specifications of the MR production rates, which significantly influence the evolution of the system. Thus, the dataset contains variation due to biological system noise within collections of cells of the same type and due to different cell types. Ultimately, we generate single-cell samples collected from five to ten cell types (Dibaeinia and Sinha, 2020).

In the second stage, the clean gene expressions sampled previously are corrupted with technical measurement error that resembles the noise phenomena found in real scRNA-seq data:

- *outlier genes*: a small set of genes have unusually high expression across measurements
- *library size*: different cells have different total UMI counts, following a log-normal distribution
- *dropouts*: a high percentage of genes are recorded with zero expression in a given measurement
- *unique molecule identifier (UMI) counts*: we observe Poisson-distributed count data rather than the clean expression values

To configure these noise modules, we use the parameters calibrated by Dibaeinia and Sinha (2020) for datasets from different scRNA-seq technologies. We extend SERGIO to allow for the generation of knockout intervention experiments. For this, we force the production rate of knocked-out genes to zero during simulation. Our implementation uses the public source code by (Dibaeinia and Sinha, 2020), which is available under a GNU General Public License v3.0.[2]

**Parameters**    Given a causal graph $G$, the parameters SERGIO requires to simulate $c$ cell types of $d$ genes are:

- $k \in \mathbb{R}^{d \times d}$: interaction strengths (only used if edge $i \to j$ exists in $G$)
- $b \in \mathbb{R}_+^{d \times c}$: MR production rates (only used if gene $j$ is a source node in $G$)
- $\gamma \in \mathbb{R}_+^{d \times d}$: Hill function coefficients controlling nonlinearity of interactions
- $\lambda \in \mathbb{R}^d$: decay rates per gene
- $\zeta \in \mathbb{R}_+^d$: scales of stochastic process noise per gene for chemical Langevin equations

The technical noise components are configured by:

- $p_{\text{outlier}} \in [0, 1]$: probability that a gene is an outlier gene
- $\mu_{\text{outlier}} \in \mathbb{R}_+, \sigma_{\text{outlier}} \in \mathbb{R}_+$: parameters of the log-normal distribution for the outlier multipliers
- $\mu_{\text{lib}} \in \mathbb{R}_+, \sigma_{\text{lib}} \in \mathbb{R}_+$: parameters of the log-normal distribution for the library size multipliers
- $\delta \in [0, 100], \xi \in \mathbb{R}_+$: dropout percentile and temperature of the logistic function parameterizing the dropout probability of a recorded expression

In our experiments, the simulator parameters are selected in the ranges suggested by Dibaeinia and Sinha (2020).

**Standardization**    There are several ways to preprocess and normalize single-cell transcriptomic data for downstream use (Robinson et al., 2010). For simplicity, we employ $\log_2$ counts-per-million (CPM) normalization, which normalizes the total UMI counts per sample and then $\log_2$-transforms the relative count values. Specifically, the CPM value for gene $j$ in sample $i$ is defined as

$$x_j^{\text{cpm},i} := \frac{x_j^i \cdot 10^6}{l_i} \quad \text{with library size } l_i = \sum_{j=1}^d x_j^i. \tag{14}$$

For zero expressions $x_j^i$, the $\log_2$-CPM values are imputed with zero. The remaining $\log_2$-CPM values range between 10 and 19, so we shift and scale the values before performing causal discovery. To replicate the sparsity pattern and the relative ordering of values within samples in the original dataset $D$, we standardize the nonzero $\log_2$-CPM values by subtracting the minimum (instead of the mean) and dividing by the overall standard deviation. All methods considered in Section 6, including AVICI, work with GRN data in this standardized $\log_2$-CPM format.

---

[2] https://github.com/PayamDiba/SERGIO

# B   Evaluation Metrics

We report several metrics to assess how well the predicted causal structures reflect the ground-truth graph. We measure the overall accuracy of the predictions and how well-calibrated the estimated uncertainties in the edge predictions are, since AVICI predicts marginal probabilities $q(g_{i,j}; \theta_{i,j})$ for every edge. Unless evaluating these edge probabilities, we use a decision threshold of $0.5$ to convert the AVICI prediction to a single graph $G$.

**Structural and edge accuracy**   The structural hamming distance (SHD) (Tsamardinos et al., 2006) reflects the graph edit distance between two graphs, i.e., the edge changes required to transform $G$ into $G'$. By contrast, the structural intervention distance (SID) (Peters and Bühlmann, 2015) quantifies the closeness of two DAGs in terms of their valid adjustment sets, which more closely resembles our intentions of using the inferred graph for downstream causal inference tasks.

SHD and SID capture global and structural similarity to the ground truth, but notions like precision and recall at the edge level are not captured well. SID is zero if and only if the true DAG is a subgraph of the predicted graph, which can reward dense predictions (Prop. 8 by Peters and Bühlmann (2015): $\mathrm{SID}(G, G') = 0$ when $G$ is empty and $G'$ is fully connected). Conversely, the trivial prediction of an empty graph achieves highly competitive SHD scores for sparse graphs.

For this reason, we report additional metrics that quantify both the trade-off between precision and recall of edges as well as the calibration of their uncertainty estimates. Specifically, given the binary predictions for all $d^2$ possible edges in the graph $G$, we compute the edge precision, edge recall, and their harmonic mean (F1-score) for each test case and estimate their means and standard errors across the test cases. Since the F1-score is high only when precision and recall are high, both empty and dense predictions are penalized and no trivial prediction scores well, making it a reliable metric for structure learning.

**Edge confidence**   To evaluate the edge probabilities predicted by AVICI and the baselines, we compute the areas under the precision-recall curve (AUPRC) and receiver operating characteristic (AUROC) when converting the probabilities into binary predictions using varying decision thresholds (Friedman and Koller, 2003). Both statistics capture different aspects of the confidence estimates. The AUROC is insensitive to changes in class imbalance (edge vs. no-edge) for a given $d$. However, when the number of variables $d$ in sparse graphs of $O(d)$ edges increases, AUROC increasingly discounts the accuracy on the shrinking proportion of edges present in the ground truth, which makes AUPRC more suitable for comparisons ranging over different $d$. The AUROC is equivalent to the probability that the method ranks a randomly chosen positive instance (i.e., an edge $i \rightarrow j$ present in the ground truth) higher than a randomly chosen negative instance (i.e., an edge $i \ldots j$ absent in the ground truth) (Fawcett, 2004).

**Calibration**   To assess the true correctness likelihood implied by the predicted edge probabilities, we use the concept of calibration (DeGroot and Fienberg, 1983; Guo et al., 2017). A classifier is said to be calibrated if a predicted edge probability of $\hat{p}_{i,j}$ empirically results in the observation of an edge in $(\hat{p}_{i,j} \times 100)\%$ of the cases, i.e.,

$$\mathbb{P}\big(g_{i,j} = 1 \,|\, \hat{p}_{i,j} = p\big) = p\,. \tag{15}$$

Following Guo et al. (2017), we can estimate the degree to which this property is satisfied for the predicted probabilities by defining $M$ intervals $I_m = \left(\frac{m-1}{M}, \frac{m}{M}\right)$ and binning all instances $i, j$ where $\hat{p}_{i,j} \in I_m$ into a set $S_m$. The empirical confidence and accuracy per bin $S_m$ are then defined as

$$\text{predicted } \hat{p}(S_m) = \frac{1}{|S_m|} \sum_{i,j \in S_m} \hat{p}_{i,j} \qquad \text{empirical } p(S_m) = \frac{1}{|S_m|} \sum_{i,j \in S_m} g_{i,j} \tag{16}$$

where a calibrated classifier has predicted $\hat{p}(S_m)$ = empirical $p(S_m)$, analogous to (15). Thus, a calibrated edge classifier induces a diagonal line when plotting the empirical $p(S_m)$ against the predicted $\hat{p}(S_m)$. The expected calibration error (ECE) is a scalar summary of this calibration plot and amounts to the weighted average of the vertical deviation from the perfect calibration line, i.e.,

$$\mathrm{ECE} = \sum_{m=1}^{M} \frac{|S_m|}{n} \left|\text{empirical } p(S_m) - \text{predicted } \hat{p}(S_m)\right| \tag{17}$$

where $n$ is the total number of evaluated samples (i.e., edges). The ECE does not capture accuracy in the sense of being able to predict all classes with high certainty, for which the other metrics are more suitable, but rather whether predicted probabilities are reflective empirical likelihood (Guo et al., 2017). In this work, we use $M = 10$ bins to compute the calibration plot lines and the ECE. The plotted calibration lines compute the calibration statistics in aggregate over all test cases to reduce the variance of the empirical counts within the bins, thus not showing standard errors.

## C    Inference Model Details

### C.1    Optimization

**Batch sizes**    Each AVICI model is trained as described in Algorithm 1. The objective $\mathcal{L}(\phi)$ relies on samples from the domain distribution $p(G, D)$ to perform Monte Carlo estimation of the expectations. During training, the number of variables $d$ in the simulated systems are chosen randomly from

$$d \in \{2, 5, 10, 20, 30, 40, 50\} \tag{18}$$

The datasets $D$ in the training distributions always have $n = 200$ samples, where with probability 0.5, the observations in a given dataset contain 50 interventional samples. The dimensionality of these training instances $G, D$ varies significantly with the number of variables $d$ and, therefore, so do the memory requirements of the forward passes of the inference model $f_\phi$.

Given these differences in problem size, we make efficient use of the GPU resources during training by performing individual primal updates in Algorithm 1 using only training instances $(G, D)$ with exactly $d$ variables, where $d$ is randomly sampled in each update step. Fixing the number of observations to $n = 200$, this allows us to increase the batch size for each considered $d$ to the maximum possible given the available GPU memory (in our case ranging from batch sizes of 27 for $d = 2$ down to 6 for $d = 50$, per 24 GiB GPU device).

During training, we tune the sampling probability of a given $d$ to ensure that $f_\phi$ sees roughly the same number of training data sets for each $d$, i.e., we oversample higher $d$, for which the effective batch size per update step is smaller. We also scale $\mathcal{L}(\phi)$ by dividing by $d^2$ to ensure an approximately equal loss and hence gradient scale across the different $d$ seen at training time.

The penalty $\mathcal{F}(\phi)$ for the acyclicity constraint is estimated using the same minibatch as for $\mathcal{L}(\phi)$.

**Buffer**    Since we have access to the complete data-generating process rather than only a fixed dataset, we approximate $\mathcal{L}(\phi)$ with minibatches that are sampled uniformly randomly from a buffer, which is continually updated with fresh data from $p(G, D)$. Specifically, we initialize a first-in-first-out buffer that holds 200 pairs $(G, D)$ for each unique number of variables $d$ considered during training. A pool of asynchronous single-CPU workers then constantly generates novel training data and replaces the oldest instances in the buffer using a producer-consumer workflow. We implement this buffer using an Apache PyArrow Plasma object store (Apache Licence 2.0). During training, we used 128 CPU workers (Appendix E).

The workers balance the data generation for different buffers to ensure an equal sample-to-insert ratio across $d$, accounting for the oversampling of higher $d$ as well as the longer computation time needed for generating data $D$ of larger $d$, for instance, in the GRN domain. In addition, the dataset $D$ of each element $(G, D)$ in the buffer contains four times more observations than $n = 200$ used during training. These observations are subsampled to obtain $n = 200$ each time a given buffer element $(G, D)$ is drawn to introduce additional diversity in the training data in case buffer elements are sampled more than once.

**Parameter updates**    The primal updates of the inference model parameters $\phi$ are performed using the LAMB optimizer with a constant base learning rate $3 \cdot 10^{-5}$ and adaptive square-root scaling by the maximum effective batch size[3](You et al., 2019). Gradients with respect to $\phi$ are clipped at a global $\ell^2$ norm of one (Pascanu et al., 2013). In all three domains, we optimize $\phi$ for a total number of 300,000 primal steps, reducing the learning rate by a factor of ten after 200,000 steps.

When adding the acyclicity contraint in LINEAR and RFF, we use a dual learning rate of $\eta = 10^{-4}$ and perform a dual update every 500 primal steps. The dual learning rate $\eta$ is warmed up with a linear

---

[3]With 8 GPU devices, this corresponds to a learning rate of $3 \cdot 10^{-5} \cdot \sqrt{8 \cdot 27} \approx 4.4 \cdot 10^{-4}$ (You et al., 2019)

schedule from zero over the first 50,000 primal steps. To reduce the variance in the dual update, we use an exponential moving average of $\mathcal{F}(\phi)$ with step size $10^{-4}$ maintained during the updates of the primal objective. To approximate the spectral radius in Eq. (8), we perform $t = 10$ power iterations initialized at $\mathbf{u}, \mathbf{v} \sim \mathcal{N}(0, \mathbf{I}_d)$.

### C.2   Architecture

As described in Section 4.2, the core of our model consists of $L = 8$ layers, each containing four residual sublayers. Different from the vanilla Transformer encoder, we employ layer normalization before each multi-head attention and feedforward module and after the last of the $L$ layers (Radford et al., 2019). The multi-head attention modules have a model size of 128, key size of 32, and 8 attention heads. The feedforward modules have a hidden size of 512 and use ReLU activations. In held-out tasks of RFF and GRN, we found that dropout in the Transformer encoder does not hurt performance in-distribution, so we increased the dropout rates from 0.0 to 0.1 and 0.3, respectively, to help generalization o.o.d. Dropout, when performed, is done before the residual layers are added, as in the vanilla Transformer (Vaswani et al., 2017).

The position-wise linear layers that map the two-dimensional representation $(\mathbf{z}^1, \ldots, \mathbf{z}^d) \in \mathbb{R}^{d \times k}$ to $\mathbf{u}^i$ and $\mathbf{v}^i$, respectively, apply layer normalization prior to their transformations. We use Kaiming uniform initialization for the weights (He et al., 2015a). The bias term inside the logistic function of Eq. (6) is initialized at $-3$ and learned alongside all other parameters $\phi$. Likewise, the scale parameter $\tau$ is learned but optimized in $\log$ space to ensure positivity, i.e., $\tau = \exp(\tau_{\log})$ where $\tau_{\log}$ is updated as part of $\phi$ and initialized at 2. When optimizing models under the acyclicity constraint, we ignore the diagonal predictions $\theta_{ii}$ and mask the corresponding loss terms.

We implement AVICI with Haiku in JAX (Hennigan et al., 2020; Bradbury et al., 2018). We converged to the above optimization and architecture specifications through experimentation on held-out instances from the training distributions $p(D)$, i.e., in-distribution.

## D   Baselines

**Algorithms and hyperparameter tuning**   We calibrate important hyperparameters for all methods on held-out problem instances from the test data distributions $\tilde{p}(D)$ of LINEAR, RFF and GRN, individually in each domain. For the following algorithms, we search over the parameters relevant for controlling sparsity and the complexity of variable interactions:

- DCDI (Brouillard et al., 2020): sparsity regularizer $\lambda \in \{10^{-2}, 10^{-1}, 1\}$, size of hidden layer in MLPs modeling the conditional distributions $\in \{8, 32\}$
- DAG-GNN (Yu et al., 2019): graph thresholding parameter $\in \{0.1, 0.2, 0.3\}$, size of hidden layer in MLP encoder and decoder $\in \{16, 64\}$
- DiBS (Lorch et al., 2021): latent kernel length scale $\ell_z \in \{3, 10, 30\}$, ... with BGe marginal likelihood (LINEAR): effective sample size (sparsity) $\alpha_\mu^{\mathrm{BGe}} \in \{0.1, 1.0\}$ ... with nonlinear Gaussian likelihood (RFF, GRN): parameter length scale $\ell_\theta \in \{30, 300, 3000\}$
- GraN-DAG (Lachapelle et al., 2020): preliminary neighborhood selection threshold $\in \{0.5, 2\}$, size of hidden layer $\in \{8, 32\}$, pruning cutoff $\in \{10^{-3}, 10^{-5}\}$
- IGSP (Wang et al., 2017): significance $\alpha \in \{10^{-2}, 10^{-3}, 10^{-4}\}$, CI test $\in \{\text{Gaussian, HSIC-}\gamma\}$
- PC (Spirtes et al., 2000): significance $\alpha \in \{10^{-2}, 10^{-3}, 10^{-4}\}$, CI test $\in \{\text{Gaussian, HSIC-}\gamma\}$

DAG-GNN, DCDI, DiBS, and GraN-DAG use 80% of the available data to perform inference and compute held-out log likelihood or ELBO scores on the other 20% of the data. The best hyperparameters are then selected by averaging the metric over five held-out instances of $d = 30$ variables. DiBS draws 10 samples from $p(G \mid D)$ using the interventional BGe score for LINEAR and a nonlinear Gaussian interventional likelihood with MLP means for RFF and GRN. DiBS assumes an observation noise of 1, uses a scale-free graph prior, and anneals acyclicty and relaxation parameters with rate 1. All remaining parameters are kept at the settings suggested by the authors.

For the PC algorithm and IGSP, there is no held-out score, so we compute the SID and F1 scores using the ground-truth causal graphs to select their optimal parameters. This would not be possible

in practice and thus favors these methods. The HSIC-$\gamma$ CI test did not scale to $d = 100$ variables, so in these cases PC and IGSP always use the Gaussian CI test. For GRN $d = 30$, IGSP also uses the Gaussian CI test because it OOMs at 100GB when using HSIC-$\gamma$. GES and GIES use the linear Gaussian BIC score function and thus do not require calibrating a sparsity parameter (Chickering, 2003; Hauser and Bühlmann, 2012). LiNGAM is based on independent component analysis and requires no regularization tuning either (Shimizu et al., 2006).

**DAG bootstrap** To estimate edge probabilities for the non-Bayesian methods in Section 6.2, we use the nonparametric DAG bootstrap (Friedman et al., 1999). We bootstrap ten datasets $D'$ from $D$ by sampling with replacement and then run each baseline individually on each bootstrapped dataset $D'$. The nonparametric probability estimate for an edge then amounts to the proportion of predicted graphs $G'$ that contain the edge.

**Implementation** For GES, GIES, PC, and LiNGAM, we run the original R implementations of the authors using an extended version of the software by Kalainathan et al. (2020) (MIT Licence). For DCDI, DAG-GNN, GraN-DAG, and DiBS, we use the Python implementations provided by the authors (Brouillard et al., 2020; Yu et al., 2019; Lachapelle et al., 2020; Lorch et al., 2021) (MIT License, Apache License 2.0, MIT License, MIT Licence). For IGSP, we use the implementation provided as part of the CausalDAG package (Squires et al., 2018) (3-Clause BSD license).

LiNGAM relies on the inversion of a covariance matrix, which frequently fails in the GRN domain due to the high sparsity in $D$. Thus, to benchmark LiNGAM in GRN, we add small Gaussian noise to the standardized count matrix $D$. For the IGSP and PC algorithms, the same numerical adjustment is needed to avoid crashes in the CI tests on GRN. Single IGSP runs that still failed for $d = 100$ were ignored when computing the metrics. In the GRN results, we ignored a small number of single runs of DCDI for $d = 100$ and PC for $d = 30$ that failed to terminate after 24 hours walltime (on a GPU machine for the former). Lastly, the CAM pruning post-processing procedure of the author's implementation of GraN-DAG (Lachapelle et al., 2020) crashes in a few instances. We skip the post-processing step in these cases.

# E Extended Results

**Compute Resources** To carry out the experiments in this work, we trained three main AVICI models and several ablations. Each model was optimized for approximately four days using 8 Quadro RTX 6000 or NVIDIA GeForce RTX 3090 GPUs (24 GiB memory each) and 128 CPUs. To perform the benchmarking experiments, all baselines were run on four to eight CPUs each for up to 24 hours, depending on the method. DCDI required one GPU to ensure a computation time of less than one day per task instance. In all experiments, test-time inference with AVICI is done on eight CPUs and no GPU.

## E.1 AVICI generalization between LINEAR and RFF

In this section, we provide additional out-of-distribution generalization results for AVICI. Specifically, we test the AVICI model trained on the LINEAR domain on inference from RFF data, and vice versa. This means that the AVICI models not only operate under distributional shifts on the parameters of their respective data-generating processes, but also on the function classes of causal mechanisms themselves. The models infer causal structure from data generated from function classes never seen during training. As in all empirical analyses of Section 6, the graph and noise parameters are additionally o.o.d., that is, the LINEAR AVICI model is tested on the o.o.d. RFF data, and vice versa.

Table 5 summarizes the results. Even under this distributional shift, the performance of both AVICI models decreases reasonably and remains on par with most baselines (Table 1). On LINEAR data, the baselines achieve F1 scores of 0.15 - 0.54 with observational and 0.33 - 0.74 with interventional data, similar to the RFF AVICI model with 0.19 and 0.45, respectively. Conversely, on RFF data, the baselines achieve F1 scores of 0.22 - 0.42 with observational and 0.34 - 0.41 with interventional data, which is also matched by the LINEAR AVICI model here with 0.27 and 0.42, respectively. Overall, the LINEAR AVICI model generalizes marginally better to RFF data as vice versa. We do not report the SID here because the R code of Peters and Bühlmann (2015) runs out of memory.

**Table 5: Generalizing from LINEAR to RFF and vice versa** ($d = 30$ **variables**). Mean SHD ($\downarrow$), F1 score ($\uparrow$), AUROC ($\uparrow$), and AUPRC ($\uparrow$) with standard error on 30 random task instances. The domain in parentheses indicates the training domain, and the header indicates the test domain. We highlight the rows in which models were evaluated on the same function class as during training, though as in all benchmarking experiments, all test datasets $D$ are sampled from the o.o.d. data-generating distributions. The metrics of the baselines corresponding to these experiments are given in Table 1.

| | LINEAR | | | |
|---|---|---|---|---|
| | SHD | F1 | AUROC | AUPRC |
| **AVICI** (trained on LINEAR) [†] | **18.9** (2.1) | **0.672** (0.04) | **0.978** (0.00) | **0.790** (0.03) |
| **AVICI** (trained on RFF) [†] | 93.4 (20.1) | 0.191 (0.03) | 0.686 (0.03) | 0.179 (0.02) |
| **AVICI** (trained on LINEAR) | **13.2** (1.8) | **0.819** (0.02) | **0.988** (0.00) | **0.892** (0.02) |
| **AVICI** (trained on RFF) | 63.6 (14.5) | 0.452 (0.05) | 0.802 (0.03) | 0.469 (0.06) |
| | RFF | | | |
| | SHD | F1 | AUROC | AUPRC |
| **AVICI** (trained on LINEAR) [†] | 36.4 (3.2) | 0.273 (0.04) | 0.784 (0.03) | 0.385 (0.04) |
| **AVICI** (trained on RFF) [†] | **21.6** (3.5) | **0.618** (0.06) | **0.854** (0.03) | **0.659** (0.06) |
| **AVICI** (trained on LINEAR) | 34.3 (3.6) | 0.420 (0.05) | 0.811 (0.03) | 0.495 (0.05) |
| **AVICI** (trained on RFF) | **18.0** (3.6) | **0.707** (0.06) | **0.888** (0.03) | **0.739** (0.06) |

[†] Only using observational data.

**Table 6: In-distribution benchmarking results** ($d = 30$ **variables**). Mean SID ($\downarrow$) and F1 score ($\uparrow$) with standard error of all methods on 30 random task instances. Methods in the top section use only observational data, in the bottom section both observational and interventional data. The best results of each section are highlighted together with those inside its 95% confidence interval according to an unequal variances $t$-test.

| | LINEAR | | RFF | | GRN | |
|---|---|---|---|---|---|---|
| Algorithm | SID | F1 | SID | F1 | SID | F1 |
| **GES** | **217.5** (38.3) | 0.643 (0.04) | 296.9 (42.6) | 0.428 (0.03) | **535.3** (44.8) | **0.147** (0.01) |
| **LiNGAM** | 500.8 (43.5) | 0.161 (0.03) | 406.0 (42.3) | 0.237 (0.02) | **590.7** (41.8) | 0.110 (0.01) |
| **PC** | 383.5 (57.3) | 0.386 (0.04) | 386.7 (55.8) | 0.431 (0.03) | **579.7** (41.9) | **0.128** (0.01) |
| **DAG-GNN** | 533.2 (47.1) | 0.127 (0.02) | 386.1 (38.7) | 0.252 (0.02) | **576.3** (43.2) | **0.159** (0.02) |
| **GraN-DAG** | 417.2 (54.1) | 0.249 (0.02) | 312.0 (38.9) | 0.477 (0.02) | **572.6** (44.9) | 0.079 (0.01) |
| **AVICI** (ours) | **178.2** (36.9) | **0.828** (0.03) | **137.8** (29.1) | **0.838** (0.02) | 607.4 (46.2) | 0.064 (0.02) |
| **GIES** | **16.5** (10.0) | **0.942** (0.01) | 290.4 (40.8) | 0.389 (0.02) | **520.4** (45.9) | 0.162 (0.02) |
| **IGSP** | 277.6 (36.9) | 0.512 (0.04) | 386.5 (44.7) | 0.391 (0.03) | **591.9** (46.1) | 0.113 (0.01) |
| **DCDI** | 242.7 (36.7) | 0.559 (0.03) | 152.4 (21.6) | 0.555 (0.02) | 624.4 (38.7) | 0.080 (0.01) |
| **AVICI** (ours) | 72.4 (20.7) | **0.948** (0.01) | **67.4** (16.8) | **0.927** (0.01) | 466.5 (49.7) | **0.316** (0.05) |

## E.2   In-Distribution Benchmarking Results for $d = 30$

Table 6 gives the benchmarking results for in-distribution data of $d = 30$ variables given the otherwise unchanged setup of Section 6.2. Contrary to the o.o.d. setting, the data is generated under homogeneous, additive noise and the parameters of their generative processes are sampled from the training domains of AVICI (cf. Table 3). However, as throughout all experiments, the datasets and its data-generating parameters themselves are unique and have not been used by AVICI during training.

Compared to the o.o.d. setting, most baselines perform roughly the same. Since the data-generating processes are sampled from its training distribution, AVICI significantly improves by moving to the easier in-distribution setting, in particular in the SCM domains, which are less noisy. In the GRN

**Table 7: Benchmarking results ($d = 100$ variables).** Mean SID ($\downarrow$) and F1 score ($\uparrow$) with standard error of all methods on 30 random task instances. Methods in the top section use only observational data, in the bottom section both observational and interventional data. We highlight the best result of each section and those within its 95% confidence interval according to an unequal variances $t$-test.

| | LINEAR | | RFF | | GRN | |
|---|---|---|---|---|---|---|
| Algorithm | SID | F1 | SID | F1 | SID | F1 |
| **GES** | **2724.6** (362.2) | **0.471** (0.02) | 4703.3 (520.9) | 0.240 (0.02) | 6787.4 (351.8) | **0.031** (0.00) |
| **LiNGAM** | 6051.7 (585.1) | 0.150 (0.01) | 5489.9 (595.7) | 0.177 (0.02) | 6726.5 (444.2) | 0.011 (0.00) |
| **PC** | 5114.1 (621.1) | 0.287 (0.02) | 5294.8 (599.0) | 0.248 (0.03) | 6894.4 (426.5) | 0.029 (0.00) |
| **DAG-GNN** | 6215.6 (598.9) | 0.101 (0.01) | 5445.5 (588.8) | 0.198 (0.02) | 6574.3 (463.8) | 0.036 (0.01) |
| **GraN-DAG** | 5307.5 (661.1) | 0.161 (0.02) | 4522.0 (581.6) | **0.421** (0.04) | 6774.8 (419.3) | 0.026 (0.01) |
| **AVICI** (ours) | 3213.8 (380.0) | **0.474** (0.04) | **3531.0** (498.0) | **0.506** (0.05) | **6661.2** (464.3) | 0.000 (0.00) |
| **GIES** | **1720.7** (306.1) | **0.639** (0.02) | 4528.3 (521.1) | 0.257 (0.03) | 6691.2 (376.9) | 0.034 (0.00) |
| **IGSP** | 4181.7 (478.3) | 0.316 (0.02) | 5544.7 (572.4) | 0.182 (0.02) | 6662.3 (401.4) | 0.027 (0.00) |
| **DCDI** | 5116.9 (525.4) | 0.105 (0.01) | 3835.1 (413.3) | 0.048 (0.00) | **4410.8** (285.0) | 0.027 (0.00) |
| **AVICI** (ours) | 2825.3 (379.0) | **0.601** (0.03) | **3231.2** (500.2) | **0.550** (0.05) | 5237.5 (469.0) | **0.172** (0.05) |

domain, some baselines achieve slightly better F1 scores compared to the o.o.d. setting, which is most likely explained by a change in the graph rather than the simulator parameter distribution, since there is no reason to believe that different generative parameters are more challenging to the baselines.

### E.3 Benchmarking Results for $d = 100$

Table 7 shows the benchmarking results for $d = 100$ variables given $n = 1000$ observations and the experimental setup of Section 6.2. We highlight that in this evaluation regime, AVICI operates under distribution shift in terms of the causal structures, mechanisms or simulator parameters, and noise distributions, as well as the number of variables and the number of observations seen during training.

Overall, the qualitative ranking of the methods is very similar as for $d = 30$. AVICI outperforms all baselines in the nonlinear RFF domain, with and without access to interventional data. Likewise, AVICI is the only method to achieve nontrivial edge accuracy in terms of F1 score on the challenging GRN domain. On the simpler LINEAR domain, there is no statistically significant difference between GES/GIES and AVICI, which perform overall most favorably.

### E.4 Benchmarking Results on Real-World Proteomics Data

We additionaly evaluate all of the methods on the real-world dataset by Sachs et al. (2005), which contains continuous measurements of $d = 11$ proteins involved in human immune system cells. Structure learning algorithms are commonly compared on this dataset, and for completeness, we report the performance of AVICI and the baselines here. However, the ground-truth network of 17 edges put forward by Sachs et al. (2005) has been challenged by some experts (Mooij et al., 2020) and the assumptions of causal sufficiency and acyclicity may not be justified even though assumed by most methods, which should be kept in mind when interpreting the results. A large part of the data are interventional, in which the measured proteins were activated or inhibited using specific reagents. Most interventions are likely not perfect and the intervention targets may not be completely accurate (Mooij et al., 2020).

For this experiment, we follow Wang et al. (2017) and Brouillard et al. (2020) and discard data in which interventions were not targeted directly at one of $d = 11$ measured proteins. Given this setup, we have $n = 5846$ data points that contain 1755 observational and 4091 interventional measurements, which consist of five single-protein perturbations. In our results, methods that only use observational data take the concatenation of all of the data without the intervention target information as input. All baselines use the hyperparameters tuned for the nonlinear RFF domain. The data is standardized to have mean 0 and variance 1.

Table 8 summarizes the results of all methods with respect to the reference causal graph. Figure 5 visualizes the prediction of each method. Overall, the results are not very conclusive. GES and GIES

**Table 8: Benchmarking results on the proteomics data by Sachs et al. (2005).** We report the SHD (↓), SID (↓), and F1 score (↑), and the number of edges predicted for all methods. Methods in the bottom section use the observational and interventional data, while the top row uses the concatenation of both, without the intervention targets. We highlight the best result of each section.

|  | SHD | SID | F1 | no. edges |
|---|---|---|---|---|
| **GES** | 35 | **44** | 0.281 | 40 |
| **LiNGAM** | 18 | 58 | 0.083 | 7 |
| **PC** | 21 | 47 | 0.244 | 24 |
| **DAG-GNN** | 26 | 49 | 0.273 | 27 |
| **GraN-DAG** | **16** | 38 | **0.473** | 21 |
| **AVICI** (ours, trained on LINEAR)$^{\parallel}$ | 20 | 56 | 0.143 | 12 |
| **AVICI** (ours, trained on RFF)$^{\parallel}$ | 17 | 56 | 0.276 | 13 |
| **GIES** | 40 | **30** | 0.286 | 46 |
| **IGSP** | 19 | 49 | 0.286 | 18 |
| **DCDI** | **15** | 42 | **0.308** | 9 |
| **AVICI** (ours, trained on LINEAR)$^{\parallel}$ | 20 | 50 | 0.250 | 11 |
| **AVICI** (ours, trained on RFF)$^{\parallel}$ | 16 | 49 | 0.267 | 15 |

$^{\parallel}$ Point estimate using decision threshold 0.5

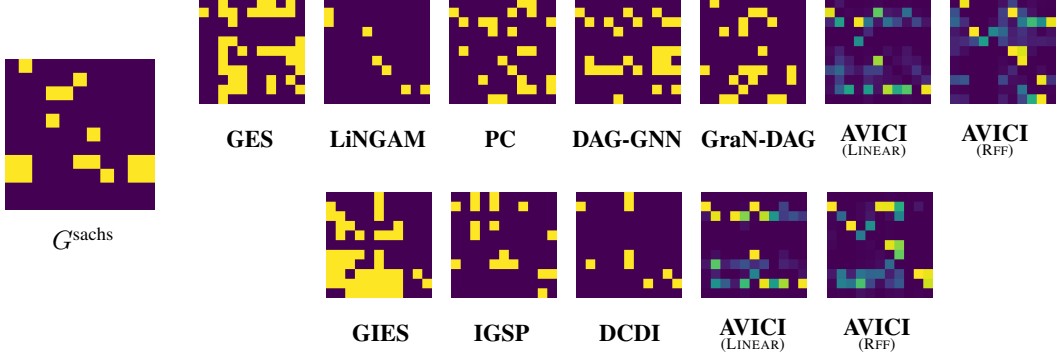

$G^{\text{sachs}}$

GES LiNGAM PC DAG-GNN GraN-DAG AVICI (LINEAR) AVICI (RFF)

GIES IGSP DCDI AVICI (LINEAR) AVICI (RFF)

**Figure 5: Prediction of each method on the proteomics dataset by Sachs et al. (2005).** All baselines predict a point estimate of $G$, where edges are painted yellow. The posterior edge probabilities predicted by AVICI, which were thresholded for Table 8, are visualized as color gradients. The bottom row of methods use observational and interventional data, while the top row only uses the concatenation of both, without the intervention targets. The believed ground truth graph is shown on the left.

perform best in terms of SID, GraN-DAG is most favorable in terms of F1, and together with DCDI and AVICI also in terms of SHD. More generally, the number of predicted edges varies greatly across methods. Almost all F1 scores fall between 0.25 and 0.30.

## E.5 Uncertainty quantification for $d = 30$

**Calibration** Figure 6 gives the calibration plots for all methods considered in the uncertainty analysis of Section 6.2 of the main text. In the SCM domains, AVICI closely traces the diagonal calibration line, both when given access to observational and mixed data. Here, the nonparametric bootstraps of the PC, GIES, and IGSP algorithms as well as DiBS are similarly well-calibrated. These baselines achieve worse expected calibration error (ECE) than AVICI because a significantly larger total proportion of AVICI's predictions are well-calibrated (cf. Equation 17). DCDI, LiNGAM, and DAG-GNN are highly overconfident, that is, they predict edges with high probability when empirically only few edges exist.

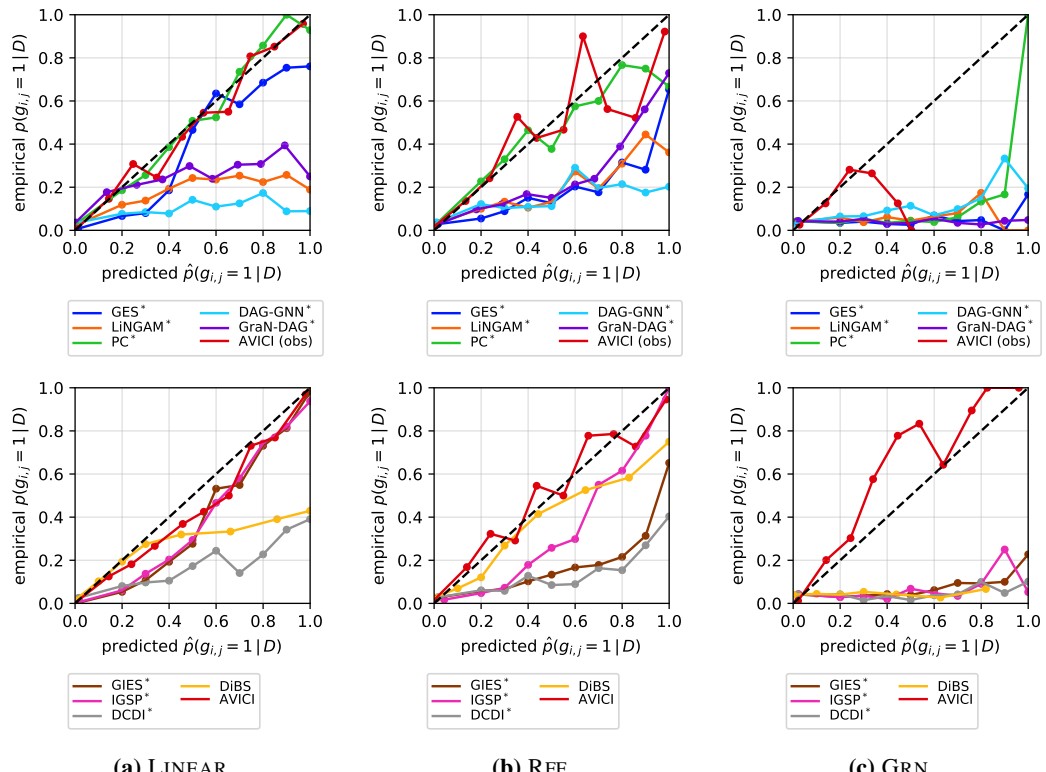

**(a)** LINEAR        **(b)** RFF        **(c)** GRN

**Figure 6: Calibration plots** ($d = 30$) for all methods in the experiments of Section 6.2 and Table 4b. The top and bottom rows of plots show the methods that use observational data and a mix of observational and interventional data, respectively. Methods with an asterix use the nonparametric DAG bootstrap to estimate edge probabilities (Friedman et al., 1999) (cf. Section 6.2).

**Probabilistic metrics**    Table 9 summarizes the probabilistic AUROC and AUPRC metrics for all methods in the experiment of Figure 4. Explanations and interpretations for both metrics are given in Section B. The relative performance of the bootstrap baselines and AVICI is similar to the point estimate benchmark. Overall, AVICI performs favorably across the three domains, with GES and GIES on par in LINEAR. However, since AUROC and AUPRC metrics evaluate the full spectrum of decision thresholds, we additionally see that AVICI achieves nontrivial accuracy in GRN even without access to gene knockout data, indicating that AVICI may provide useful information even in settings where only passive observations are available. This aspect is not apparent when converting the posterior probability estimates of AVICI based on a single threshold and then comparing SID and F1 scores as in Table 1.

**Table 9: Probabilistic metrics for the benchmark ($d = 30$ variables).** Mean AUROC ($\uparrow$) and AUPRC ($\uparrow$) with standard error of all methods on ten random task instances. Methods in the top section use only observational data, in the bottom section both observational and interventional data. We highlight the best result of each section and those within its 95% confidence interval according to an unequal variances $t$-test.

| Algorithm | LINEAR | | RFF | | GRN | |
|---|---|---|---|---|---|---|
| | AUROC | AUPRC | AUROC | AUPRC | AUROC | AUPRC |
| **GES**[*] | 0.930 (0.01) | **0.643** (0.05) | **0.759** (0.04) | 0.289 (0.06) | **0.496** (0.02) | 0.045 (0.00) |
| **LiNGAM**[*] | 0.752 (0.06) | 0.365 (0.10) | 0.701 (0.04) | 0.229 (0.03) | **0.537** (0.03) | 0.057 (0.01) |
| **PC**[*] | 0.771 (0.04) | 0.469 (0.06) | **0.825** (0.04) | **0.507** (0.07) | **0.510** (0.02) | 0.052 (0.01) |
| **DAG-GNN**[*] | 0.621 (0.03) | 0.097 (0.02) | 0.693 (0.03) | 0.174 (0.01) | **0.547** (0.05) | 0.082 (0.03) |
| **GraN-DAG**[*] | 0.685 (0.04) | 0.222 (0.03) | 0.781 (0.04) | 0.419 (0.09) | **0.534** (0.08) | **0.113** (0.05) |
| **AVICI** (ours) | **0.979** (0.01) | **0.767** (0.06) | **0.801** (0.07) | **0.571** (0.12) | **0.678** (0.10) | **0.185** (0.04) |
| **GIES**[*] | **0.981** (0.01) | **0.879** (0.04) | **0.769** (0.06) | 0.389 (0.08) | 0.517 (0.03) | 0.070 (0.01) |
| **IGSP**[*] | 0.942 (0.01) | 0.660 (0.05) | **0.822** (0.04) | 0.374 (0.07) | 0.471 (0.02) | 0.049 (0.01) |
| **DCDI**[*] | 0.771 (0.03) | 0.306 (0.03) | **0.740** (0.05) | 0.283 (0.06) | 0.557 (0.07) | 0.113 (0.05) |
| **DiBS** | 0.837 (0.03) | 0.524 (0.05) | **0.740** (0.05) | 0.340 (0.05) | 0.517 (0.03) | 0.048 (0.01) |
| **AVICI** (ours) | **0.987** (0.01) | **0.902** (0.04) | **0.862** (0.05) | **0.669** (0.10) | **0.901** (0.04) | **0.656** (0.10) |

[*] Nonparametric DAG bootstrap (Friedman et al., 1999)