# OpenReview forum: "Amortized Inference for Causal Structure Learning"
_NeurIPS.cc/2022/Conference — NeurIPS 2022 Accept_

### Official Review · Reviewer_5Lgh · 2022-07-05

**Rating:** 6
**Confidence:** 3
**Soundness:** 3 good
**Presentation:** 4 excellent
**Contribution:** 3 good

**Summary:**

The paper introduces an amortized inference approach to learn causal structures from observational/interventional data, by leveraging variational inference method and permutation invariant/equivarient models. Experiment results on synthetic and semi-synthetic data are provided to show that the proposed approach exhibits generalization capabilities under distribution shifts.

**Questions:**

The suggestions and questions for the authors are outlined in my comment to the previous question.

**Limitations:**

The authors have adequately addressed the limitations and potential negative societal impact of their work.

**Strengths And Weaknesses:**

Pros:
- The proposed method is sound and novel.
- The problem studied is interesting and relevant to the field of causal discovery, which helps bridge the gap between causality and machine learning.
- The empirical studies are conducted across different data generating processes, number of variables, graph types, etc.

Cons:
- The proposed idea of amortized inference for multivariate causal discovery is not completely novel. It has been explored by existing work "Supervised Whole DAG Causal Discovery", which should be discussed in the introduction and compared in the experiments.
- The model architecture in Sec. 4.2 appears to be very similar to the encoder-decoder architecture for causal graph generation in paper "Causal Discovery with Reinforcement Learning", and should be discussed/compared in the paper. The latter also applies a transformer encoder that attends over axis $d$ and a similar decoder to generate graph. The only difference appears to be that the latter does not apply attention over axis $n$.
- It would be great to provide some ablation studies on the attention over axes $n$ and $d$ to study the contribution of each module.
- The current OOD experiments change only the linear function weights, kernel scales, and noise distribution (L224-233). I believe that the ability to generalize to a completely different data-generating process is also important, e.g., train distribution contains RFF dataset while test distribution contains MLP dataset. Another aspect is to generalize from synthetic data to real data, e.g., train distribution contains RFF dataset while test dataset is a real dataset (such as the one by "Causal Protein-Signaling Networks Derived from Multiparameter Single-Cell Data").
- Could the authors explain how users should apply the learned amortized model in practice? Will there be a separate learned model for different data-generating processes like linear, RFF, MLP, and GRN? That means users have to manually train/pick an amortized model for a dataset of interest based on their understanding of the dataset. Could we combine all different data-generating processes into a large dataset and train a single, more powerful, amortized model? This is what I might expect from a amortized/supervised causal discovery method.
- Stronger baselines:
    - DAG-GNN is only applicable for a limited type of nonlinear data-generating process (and not RFF). I suggest the authors to include GRAN-DAG ("Gradient-Based Neural DAG Learning") with preliminary neighborhood selection and pruning as a baseline.
    - The authors should also include non-parametric scoring criterion and test for GES and PC.

Minor:
- It was surprising to see that LiNGAM does not perform well on the linear non-Gaussian synthetic data. Have the authors considered the improved method in "DirectLiNGAM: A Direct Method for Learning a Linear Non-Gaussian Structural Equation Model"?
- The proposed method cannot ensure convergence to acyclic solution, which might be important in some cases. Could the authors explain their choice of dual ascent to solve the acyclicity-constrained problem, instead of penalty/augmented Lagrangian method (which is more widely adopted by existing works on differentiable causal discovery)?

Post-rebuttal:

I have read the rebuttals and further responses from the authors. I think the paper still lacks a number of experiments:
- Systematic ablation study (this is particularly important in my opinion)
- Comparison with existing decoders in Zhu et al., 2020
- Benchmarks for in-distribution setting, i.e., homogeneous noise (the proposed method should ideally perform well as compared to the baselines for both in-distribution and out-distribution settings)
- Stronger baselines like GRAN-DAG and GES using nonparametric scoring criterion

Given the limited time of rebuttal period, it is understandable that it is difficult to finish conducting all experiments, and the authors have promised to include them in the final version. Therefore, my concerns have been mostly addressed and I have increased my score to 6.

---

> ### Author Response · Authors · 2022-08-02
> **Response to Reviewer 5Lgh [1/2]**
>
> Thank you very much for your careful reading and assessment of our work as well as the suggestions. We respond to your questions and concerns individually below:
>
> **More baselines and experiments**
>
> > “I believe that the ability to generalize to a completely different data-generating process is also important, e.g., train distribution contains RFF dataset while test distribution contains MLP dataset. Another aspect is to generalize from synthetic data to real data, e.g., train distribution contains RFF dataset while test dataset is a real dataset (such as the one by [Sachs et al., 2005]).”
>
> Thank you for these suggestions. In our revision, we have added several additional results as prompted by your comment. In Appendix E.1 and Table 4, we now show results in which the AVICI model trained on linear SCMs is evaluated on test-data from nonlinear RFF SCMs, and vice versa, which should constitute an even stronger shift than from the suggestion of RFF to similar nonlinear data. As expected, the performance decreases under this strong distributional shift, though we find that AVICI still performs comparable to most baselines (cf. Table 1).
>
> In addition, we ran the AVICI models and all baselines on the real-world proteomics dataset you suggested (Sachs et al., 2005), with the results given in Appendix E.3 and Table 6 and predictions visualized in Figure 4. Here, GES/GIES, DCDI and AVICI each perform best in some metric and overall most favorably w.r.t. to the gold-standard causal graph.
>
> > “DAG-GNN is only applicable for a limited type of nonlinear data-generating process (and not RFF). I suggest the authors to include GRAN-DAG [...] as a baseline.”
>
> DAG-GNN uses a general MLP encoder and decoder to learn $G$, so it is unclear to us why the method would not be suited for learning causal structure from data generated by nonlinear RFF functions. We currently do not see sufficient reason for using additional compute to also benchmark GRAN-DAG. We do already cite GRAN-DAG as related work in line 86 of Section 2.
>
> > “The authors should also include non-parametric scoring criterion and test for GES and PC.”
>
> The CI tests for the PC algorithm contain the HSIC-$\gamma$ test, which is nonparametric (Appendix D). We followed existing work in benchmarking GES using the Gaussian BIC score. If you recommend a specific different score, we would be happy to consider it.
>
> > “It was surprising to see that LiNGAM does not perform well on the linear non-Gaussian synthetic data. [...]  Have the authors considered [...] DirectLiNGAM?”
>
> In our benchmarking experiments, we evaluate all methods and AVICI on out-of-distribution data, in which the noise is heteroscedastic. One reason why LiNGAM may not perform well could be that heteroscedasticity does not match the assumptions underlying LiNGAM, or DirectLiNGAM for that matter.
>
>  &nbsp;
>
> **Related work**
>
> > “The model architecture in Sec. 4.2 appears to be very similar to [...] "Causal Discovery with Reinforcement Learning" [...] The only difference appears to be that the latter does not apply attention over axis n. [...] It would be great to provide some ablation studies.”
>
> The mentioned work also applies a transformer over axis $n$, which is a natural choice. However, we found our attention schema over both axes in combination with the inner-product graph model to be crucial for high performance across all settings. In particular, their decoder is different and not permutation equivariant, so their overall architecture does not emulate the equivariances of ours. Their experiments do not go beyond 10-node graphs for nonlinear and 30-node graphs for linear SCMs and do not evaluate o.o.d, which is evidence that the attention and equivariances of AVICI are crucial, allowing our proposed method to scale to at least 100 nodes and o.o.d. (Figure 3).
>
> We do not have the resources to train architecture ablations of our models for this discussion period, but we are happy to perform these experiments for the camera-ready version if you suggest that we do so.
>
> > “The proposed idea [...] has been explored by existing work ‘Supervised Whole DAG Causal Discovery’”
>
> Thank you for the pointer. We were not aware of this work since it is a preprint. In their work, the authors predict $G$ given the $d \times d$ correlation coefficients among the variables. Their approach neither allows using interventional data nor the full $n \times d$ data set $D$, which makes it hard to learn anything in applications like gene regulatory networks. We have added a discussion of this work in lines 89-90 of Section 2.

---

> ### Author Response · Authors · 2022-08-02
> **Response to Reviewer 5Lgh [2/2]**
>
> **Questions**
>
> > “Could the authors explain how users should apply the learned amortized model in practice? Will there be a separate learned model for different data-generating processes [...]? Could we combine all different data-generating processes into a large dataset and train a single, more powerful, amortized model?”
>
> Users may train an AVICI model for any simulator of their domain. When using our pretrained models, users could either use the SCM-trained models for continuous-valued data or the GRN model specifically for inferring gene regulatory networks. The additional experiments we ran as prompted by your review (Appendix E.1, Table 4 and Appendix E.3, Table 6 + Figure 4) show that the models trained as part of this work perform well on real data and under very drastic distributional shift, indicating that our models will be useful for practitioners essentially out-of-the-box.
>
> If it were a user’s goal to train a single, generalist inference model, they could merge all the training distributions. However, the motivation of our work lies primarily in providing a method for specifying more realistic inductive biases for causal discovery, not in blurring the boundaries between domains indefinitely.
>
> > “Could the authors explain their choice of dual ascent to solve the acyclicity-constrained problem, instead of penalty/augmented Lagrangian method?”
>
> As you correctly point out, the augmented Lagrangian method has often been employed in continuous causal discovery. However, in this work, we _train a neural network_ to predict a variational approximation of the posterior over $G$, which is a very different constrained optimization problem in nature. For this reason, we rely on state-of-the-art methods that are specifically tailored for deep learning with constraints (Nandwani et al., 2019), which also have convergence results (Jin et al., 2020).
>
> We hope that we have addressed all of your questions and concerns and, given the additional experiments, that we were able to improve your opinion of our work for the better. Please do not hesitate to reach out for further clarifications if needed.

---

> ### Comment · Reviewer_5Lgh · 2022-08-07
> **Further Comments**
>
> Thanks for the detailed response and the revision with additional experiments. Here are some of my further comments:
> - **DAG-GNN**: For DAG-GNN, the nonlinear functions $f_1$ and $f_2$ in Eq. (3) (of DAG-GNN paper) are applied variable-wise to each of the variables (similarly in Eq. (5)). Therefore, to the best of my knowledge, it can only handle a limited type of nonlinear SEM, and certainly not general ANMs. Based on experiments in Section 4.1 of their paper, it seems that DAG-GNN is able to handle nonlinear SEM with similar form as $X=f_1(Af_2(X))+\epsilon$ with $A$ being a weighted adjacency matrix. In this case, DAG-GNN is not general enough to be applicable for RFF data. The authors could refer to Section 4.1 in the GRAN-DAG paper which shows that GRAN-DAG performs much better than DAG-GNN on the Gaussian process/RFF data (which I believe may be similar to the setup considered by the authors), since GRAN-DAG may be general enough to learn RFF. Therefore, I think using GRAN-DAG (or NOTEARS-MLP) would be a much fairer comparison.
> - **GES**: I think using a nonparametric score function such as the one in "Generalized Score Functions for Causal Discovery" might be better. It seems to me that this method has been adopted as a baseline by a number of nonparametric causal structure learning papers, such as the GRAN-DAG and NOTEARS-MLP papers.
> - **Heteroscedasticity for out-of-distribution data**: In this case, could the authors clarify in the paper whether any of the baselines can handle this kind of heteroscedasticity? It seems to me that this heteroscedasticity for o.o.d. data does not match the assumptions for most of the baselines considered in Table 1.
> - **Related work**:
>     - As the authors described, there are indeed differences (e.g. equivariances) between the mentioned work and the proposed method. However, I still think that the overall high-level architecture shares some similarities, i.e., the mentioned work also used an encoder-decoder architecture for graph generation, where the encoder is a transformer and the decoder takes in encoded representations in a pairwise fashion to output some probabilities/logits. Therefore, I think it makes sense to include such a discussion in the paper to make clear the similarities and differences.
>     - For curiosity, could the authors provide more explanation about why the decoder in the mentioned work is not permutation equivariant? Lines 176-185 state that two linear layers are used to map the encoded representations to two embeddings, and then dot product is taken w.r.t. them. From what I read, doesn't this seem similar to the decoder in the mentioned work (Section 4), except that the latter applies addition instead of dot product?
>     - "which is evidence that the attention and equivariances of AVICI are crucial, allowing our proposed method to scale to at least 100 nodes and o.o.d.": Based on my understanding, this might not be a completely fair statement. The mentioned work is using reinforcement learning, and the proposed method is using amortized variational inference. Since these are two different training strategies (and are not directly comparable in terms of scalability), their work not being able to scale up to large graphs doesn't seem to support the importance of the attention and equivariances of AVICI for better scalability.
> - **Ablation studies**: I believe that ablations studies are crucial to support the authors' claims such as "we found our attention schema over both axes in combination with the inner-product graph model to be crucial for high performance across all settings". How much does the attention over each axis contribute? How much does the permutation equivariance contribute?
> - **Choice of dual ascent**: Including such a discussion in the paper would greatly benefit readers like me, so that they could have a better understanding about the motivation of such a choice.

---

> > ### Author Response · Authors · 2022-08-08
> > **Response to Further Comments by Reviewer 5Lgh**
> >
> > Thank you very much for the further comments and the additional suggestions.
> >
> > **Suitability of baselines and experiments:**
> > In general, we would like to emphasize that the goal of our experiments is not to evaluate the baselines and AVICI on synthetic data that perfectly matches their assumptions. We view synthetic data experiments as a proxy for performance on real-world data (lines 219-224), where model assumptions are never fully met, thus evaluating methods on their exact assumptions would not be a good proxy. For causal discovery in particular, many assumptions may be unrealistic, see e.g. Reisach et al. (2021). Since it is infeasible for us to evaluate all possible experimental conditions, we opted for more challenging settings. However, we would be happy to consider benchmarking the baselines also for homogeneous noise in the final version. As shown in Figures 3a-b (top), the performance of AVICI is significantly better in this case.
> >
> > Given your response, we agree that GRAN-DAG or NOTEARS-MLP may be more well-suited for comparison on RFF than DAG-GNN. We will implement this additional baseline for the final version and correspondingly amend all of the tables. (We won’t be able to finish this until the revision deadline tomorrow). It may be relevant to highlight that we evaluated DiBS using the NOTEARS-MLP model in the RFF domain (Table 7, Appendix E), which performs very similarly to the bootstrapped nonlinear baselines.
> >
> > **Ablation studies and relation to Zhu et al., 2020** (“Causal Discovery with Reinforcement Learning")**:** We agree that a discussion of this work will be helpful to readers. In our most recent revision, we have contextualized the work in lines 86-87 and lines 164-165, and we will add additional discussion after the results of the ablation studies are finalized (see below). Overall, however, we also agree that their work is “not directly comparable [to ours] in terms of scalability” since they do not perform amortized inference.
> >
> > Following our previous response, we will perform ablations for the camera-ready version to justify our respective architecture choices better. At this stage, we do want to acknowledge that we previously experimented with a decoder similar to Zhu et al., whose decoder corresponds to a single-layer relational network [A] where $g_{ij} = \text{MLP}_\theta([z_i, z_j])$ with $[z_i, z_j]$ denoting concatenation. We tested a three-layer variant of this decoder in initial experiments (see `use_relational_net` flag in `architectures_general.py` of our code) but found performance to be worse and ultimately converged to our decoder. Thanks to your comment, we noticed that we wrongly concluded the decoder of Zhu et al. to not be permutation equivariant; we are sorry for this misunderstanding. Our ablation experiments will include an evaluation of the decoder of Zhu et al.
> >
> > **Choice of dual ascent:** Thank you for this suggestion. We have added this discussion in lines 209-214 of our most recent revision.
> >
> > We thank the reviewer for the insightful and constructive feedback. We will acknowledge the reviewer in the final version of the paper. We are available for further clarification in case there are any remaining concerns that have not been sufficiently addressed.
> >
> >
> > [A] Santoro, Adam, et al. "A simple neural network module for relational reasoning." Advances in neural information processing systems 30 (2017).

---

### Official Review · Reviewer_My1M · 2022-07-11

**Rating:** 7
**Confidence:** 3
**Soundness:** 3 good
**Presentation:** 4 excellent
**Contribution:** 3 good

**Summary:**

This paper proposes a method for learning a causal graph from data, by training a neural network on similar data from a simulator that generates both random graphs and data from those graphs, so that the model learns to predict graph structures given data. Data points can be marked as observational or interventional in arbitrary ways.

**Questions:**

1. (Section 2.1) I wasn't sure at first what definition of causal structure you meant here. But having read the paper, I suppose it doesn't really matter: whatever the relation between graph and data might be, if you have a simulator that captures this relation, AVICI will learn how to recover graphs given datasets. Or are there assumptions I'm overlooking here?

2. (Section 2.2) The sentence on lines 80-82, comparing AVICI to work leveraging multi-domain data, came across as odd. The methods just have different (orthogonal?) goals, but here it sounds like those other methods are weakened by some assumption. ("Orthogonal", because maybe if AVICI or a method like it is fed multi-domain datasets, it could learn to do the same task.)

3. (Figure 3) The SID- and SHD-ratios have a counterintuitive definition, and furthermore are apparently usually larger than 1, often by quite a bit it seems from the direction of the red and green lines at F1. Is there a better way to visualise these results?

Minor points:
* below (1): "for any" -> "for some" ($\exists$, not $\forall$)


**Limitations:**

Yes

**Strengths And Weaknesses:**

The method proposed seems extremely powerful and general, both judging by its design and by its empirical performance. I expect it will have a very high impact.

The paper is also very clearly written.

I am not familiar enough with related work on amortized variational inference to assess the originality of this approach.

UPDATE: I learned from the other reviewers that this submission is more closely related to existing work than I thought. I have reduced my score (as well as my confidence) accordingly.

---

> ### Author Response · Authors · 2022-08-02
> **Response to Reviewer My1M**
>
> We thank you very much for the effort in reviewing our work and the valuable feedback. Below, we reply in-line to each of your comments:
>
> > “I wasn't sure at first what definition of causal structure you meant here. But having read the paper, I suppose it doesn't really matter [...]”
>
> Correct, as long as the graph $G$ encodes the notion of causal structure we want to infer and the data-generating process $p(D|G)$ captures this notion given $G$, here formalized as Equation (1) (lines 66-69), AVICI will learn to infer the posterior over $G$. We used this definition to emphasize that our approach can also handle cyclic systems, not because the definition itself is crucial.
>
> > “[...] comparing AVICI to work leveraging multi-domain data came across as odd. The methods just have different (orthogonal?) goals [...]”
>
> We agree that this discussion was misleading. We removed the sentence on multi-domain methods in our revision.
>
> > “The SID- and SHD-ratios have a counterintuitive definition, and furthermore are apparently usually larger than 1, often by quite a bit it seems from the direction of the red and green lines at F1. Is there a better way to visualise these results?”
>
> We are happy to implement a different transformation or add additional bar plots (with absolute SID/SHD values) if you suggest that we do so. We proposed this transformation to make the SID/SHD scores interpretable on the radar plot in Figure 3, where larger scores correspond to better performance, like the other metrics. Since Figure 3 is about out-of-distribution generalization, we think that in-distribution SID/SHD performance is a suitable reference quantity.
>
> Finally, thank you for the correction of (1). We have corrected it in the manuscript.

---

### Official Review · Reviewer_imor · 2022-07-13

**Rating:** 6
**Confidence:** 4
**Soundness:** 3 good
**Presentation:** 4 excellent
**Contribution:** 3 good

**Summary:**

In this paper, the authors aim to address the problem of causal discovery. Considering that the existing methods for causal discovery are hard to specify realistic inductive biases and require different assumptions, the authors propose an amortized variational inference model for causal discovery, which amortizes the process of causal structure learning and approximates the posterior over causal graphs given a data-generating distribution. The authors evaluate the performance of the proposed method on the synthetic data 13 and semi-synthetic gene expression data. However, there are some concerns as follows:

**Questions:**

Please refer 'Strengths And Weaknesses'

**Limitations:**

Please refer 'Strengths And Weaknesses'

**Strengths And Weaknesses:**


1.	One of the main concerns is that the authors do not provide sufficient theoretical validation, such as in the form of identifiability or consistency proofs. Identifiability is an important problem for causal discovery from observational data, and the authors seem to ignore this problem.
2.	The main contribution of this paper is to leverage the variational inference to discover the causal structure. However, similar ideas have been proposed by other researchers like [1]. The main difference between the proposed method and [2] is that [2] is used for time-series data.  Therefore, the contributions of these are limited.
3.	The authors assume causal sufficiency, so all the data should be extracted from the same distribution with one causal mechanism. And the domain variables that result in the circumstance of domain-shift or out-of-distribution [2], however, the authors consider the OOD generalization and cannot observe the domain variables. Hence, these experiments seem to be not reasonable.
4.	Technologically, since the training data are extracted from one distribution, so only one causal structure can be learned. However, since the authors employ mini-batch optimization to learn the structure. What if different structures are extracted from different samples?
5.	Finally, it is suggested that the authors should consider more latest baselines.


[1] Amortized Causal Discovery: Learning to Infer Causal Graphs from Time-Series Data
[2] domain adaptation as a problem of inference on graphical model

---

> ### Author Response · Authors · 2022-08-02
> **Response to Reviewer imor**
>
> Thank you for your assessment of our work. We believe that there are several important misunderstandings of our approach and prior work, which we would like to clarify before addressing other concerns and questions:
>
> > “The authors assume causal sufficiency, so all the data should be extracted from the same distribution with one causal mechanism. And the domain variables that result in the circumstance of domain-shift or out-of-distribution [2], however, the authors consider the OOD generalization and cannot observe the domain variables. Hence, these experiments seem to be not reasonable.”
>
> We do not understand what is meant by this comment. As stated in lines 74-75, causal sufficiency means that there is no unobserved confounding, i.e., that we observe all common parents among $\mathbf{x}$ for a given task. Causal sufficiency does _not_ mean that all data is “extracted from the same distribution”. When we evaluate the inference model under distribution shift, the data-generating processes for the datasets $D$ change, but we still observe all causal variables $\mathbf{x}$ of $G$. Could you please provide a clarification?
>
> > “Technologically, since the training data are extracted from one distribution, so only one causal structure can be learned. However, since the authors employ mini-batch optimization to learn the structure. What if different structures are extracted from different samples?”
>
> We also do not understand what is meant here. The training distribution consists of samples $G,D \sim p(G,D)$ with _different_ causal structures and corresponding datasets, and the inference model learns to generalize to unseen $G, D$. You are right that, during training, there is “one distribution” $p(G,D)$, but each sample from this distribution is generated by first sampling a fresh graph $G$ and then sampling observations $D$ given $G$ (see lines 96-99). Optimization is done over mini-batches of these unique graph-dataset pairs $G, D$. We would need a clarification from your side to resolve any remaining confusions or concerns.
>
> > “The main contribution of this paper is to leverage the variational inference to discover the causal structure. However, similar ideas have been proposed by other researchers like [1]. The main difference between the proposed method and [2] is that [2] is used for time-series data.”
>
> The work in [1] is on domain adaptation and not very related to our work. [1] uses GANs for learning conditional distributions and a variant of CD-NOD, a structure learning algorithm, for learning a graphical model. [1] does not amortize or evaluate structure learning but instead uses notions from graphical models to improve domain adaptation.
>
> The work in [2] is already cited (Löwe et al., 2020) in line 89 of the related work section. Our approach is significantly more general than [2]. AVICI learns to infer unseen causal structures $G$ from i.i.d. data $D$ of _unseen causal mechanisms_, even generalizing to causal mechanisms from _unseen data-generating processes and real data_ (see Sections 6, E.1, E.3), and handles interventional data. By contrast, [2] assume that the causal mechanisms among the variables $\mathbf{x}$ are _the same_ in each task, allowing only $G$ to change and only operating on observational data. Thus, there are significant methodological differences beyond the fact that [2] only works for time-series data.
>
> > “Identifiability is an important problem for causal discovery from observational data, and the authors seem to ignore this problem.”
>
> Identifiability of $G$ from $p(\mathbf{x})$ is an infinite-sample property derived from assumptions on the data-generating process and not from the inference method, for example, the type of noise or function class of a postulated SCM. If we assume that $D$ originated from an identifiable model, $G$ is in theory identifiable from observational data also for AVICI. Similar to other gradient-based structure learning works, it is true that AVICI does not represent unidentifiability in the form of an MEC. However, AVICI does provide posterior uncertainties for each edge. In additional analyses of our revision, we demonstrate that these uncertainties are well-calibrated (Figures 5 and 6 in Appendix E). Beyond this, AVICI allows inference with interventional data, which can also resolve identifiability issues, but again, these are not a property of the inference method.
>
> > “It is suggested that the authors should consider more latest baselines.”
>
> We provide several recent baselines, e.g. DAG-GNN (Yu et al., 2019) and DCDI (Brouillard et al., 2020). If there are any specific additional methods you would like us to benchmark, we would be happy to add them.
>
> We hope that we have addressed your questions and concerns and that we were able to improve your opinion of our work for the better, especially since it seems that there might have been some misunderstandings. Please get back to us with clarifications and additional questions if needed.

---

> > ### Comment · Reviewer_imor · 2022-08-08
> > **Response to the authors**
> >
> >  I thank the authors for providing their feedback, they address most of my concerns. And I change the score. However, I have some questions.
> >
> > I still consider that the contribution of this paper is limited. From the perspective of model, the proposed method is similar to [1], and it is suggested that the authors should consider [1] as the compared method.
> > As for the identification theory, I think the contribution of the gradient-based is the acyclicity restriction. From the perspective of causal discovery, identification is important or it is unclear if the learned structure is unique. And it is too strong to assume that

---

> > > ### Author Response · Authors · 2022-08-08
> > > **Response to Further Comments by Reviewer imor**
> > >
> > > Thank you for getting back to us. We are glad that we addressed most of your concerns and that our explanations updated your evaluation of our work.
> > >
> > > **Relation to [1] (Löwe et al., 2022):** (In our reply, we accidentally denoted this work by [2] since you mentioned [2] in the context of time series. Apologies for this confusion).
> > >
> > > As pointed out in our previous reply, Löwe et al. (2022) solve a different and significantly more limited problem as our work. Their approach assumes that _the same data-generating process_ (i.e. the same function) generates any considered dataset $D$ given a provided causal graph $G$. Please refer to Eq. 2 of Section 3 in [Löwe et al., (2020)](https://arxiv.org/pdf/2006.10833.pdf) for details. For their setting, Löwe et al. propose a time-series model for inferring $G$ given $D$. However, the method by Löwe et al. _cannot and is not intended for_ predicting $G$ when the data is generated by different data-generating mechanisms, e.g. different weights in a linear SCM, different GP functions, or different regulatory parameters in a gene network. This, however, is the problem solved by our approach (see Section 3).
> > >
> > > Since the setting of Löwe et al. is entirely different from ours, we do not see a fair or insightful way of comparing AVICI to their approach. It would contradict our goal of amortizing causal structure learning given arbitrary datasets $D$. The fact that Löwe et al. operate with time-series data and cannot leverage interventional data are additional, secondary differences. Their model and objective are also very different and consist of the backward KL typical in variational inference, involving the likelihood reconstruction term (cf. Section 3.2), and a graph neural network.
> > >
> > > **Identifiability:** We believe that there are still some misunderstandings. Identifiability of the causal structure $G$ from the density $p(x)$ can be derived from assumptions on the data-generating process, e.g. for linear additive non-Gaussian SCMs (Shimizu et al., 2006) or nonlinear additive Gaussian SCMs (Hoyer et al., 2008). Such results show that certain data-generating processes are identifiable given the true observational density. This is not available from finite data.
> > >
> > > In this work, we are not interested in asymptotic properties of a data-generating process but in improving performance of causal structure learning from finite data. AVICI does _not_ assume any specific additive noise model but instead learns inductive bias from data. This goal is in line with many recent structure learning algorithms like NOTEARS (Zheng et al., 2018), DAG-GNN (Yu et al., 2019) or GRAN-DAG (Lachapelle et al., 2019), which do not derive identifiability results for a model class but instead propose new inference methods for structure learning. Therefore, it is not immediate that identifiability is of concern in our setting.
> > >
> > >
> > > We do not understand what the reviewer means by: _“the contribution of the gradient-based is the acyclicity restriction”_. We need further explanation here to provide a clarification.
> > >
> > > We hope that we have addressed your remaining questions and that we were able to further update your opinion of our work, in particular regarding the novelty of our contribution w.r.t. related work. Please get back to us inside the discussion period should that not be the case.

---

### Author Response · Authors · 2022-08-02
**General response to all Reviewers and the AC**

We would like to thank all of the reviewers for their effort in reading and assessing our work. We are pleased that they recognize our contribution as “sound and novel” [5Lgh], expecting “high impact” [My1M], and “extremely powerful and general, both judging by its design and by its empirical performance” [My1M], that our method is “interesting and relevant to the field of causal discovery” [5Lgh], that our paper is “very clearly written” [My1M], and that our approach “helps bridge the gap between causality and machine learning” [5Lgh].

The reviewers have some questions, mainly regarding additional experiments and the discussion of related work, as well as smaller clarifications. We address the individual questions in separate responses and have incorporated the reviewers’ suggestions in our revision.

In our revision, we added several additional results:
- As suggested by reviewer 5Lgh, we additionally evaluated the generalization ability to entirely unseen function classes of causal mechanisms by testing the AVICI model trained on linear SCM data at inference from data of nonlinear SCMs (RFF), and vice versa. The results are given in Appendix Section E.1 and Table 4. Even under this strong distributional shift, the performance of both AVICI models decreases very reasonably and remains on par with most baselines.
- As suggested by reviewer 5Lgh, we also report results on the real-world proteomics data by Sachs et al. (2005). The setup is described in Appendix E.3 and results are given in Table 6 with predictions visualized in Figure 4. We find that GES/GIES, DCDI, and AVIC perform most favorably relative to the consensus causal graph, each performing best in some metric and data setting.
- We also added additional analyses of the predicted probabilities in the form of calibration plots and expected calibration errors (Figure 5, Section 6.2), which demonstrate that the probabilities of AVICI are the most calibrated among bootstrap variants of the baselines. We kindly ask the reviewers to let us know in case they have any objections to moving this material to the main paper for the camera-ready version.

We also further polished the writing and, inspired by Radford et al. (2021), changed Eq. (6) to first $\ell_2$-normalize and then scale the node embeddings. The experimental results reflect this change in our revision, but the overall results remain qualitatively unchanged. To emphasize statistical significance, all tables now use t-tests for highlighting.

---

### Meta-Review · Area_Chair_aHCX · 2022-08-26

**Recommendation:** Accept
**Confidence:** Certain

**Metareview:**

In this paper, the authors introduce an amortized inference approach to learn causal structures from observational/interventional data. Most of the reviewers consider the proposed method is sound and novel and their questions have been well addressed.

**Award:**

No

---

### Decision · Program_Chairs · 2022-09-14

Accept